# Characterizing Human Semantic Navigation in Concept Production as Trajectories in Embedding Space

**Felipe D. Toro-Hernández**[1]    **Jesuino Vieira Filho**[2]    **Rodrigo M. Cabral-Carvalho**[1]

[1]Center of Mathematics, Computing and Cognition, Federal University of ABC
[2]Department of Computer Science and Operations Research, Université de Montréal
[1]{ftoroher,rodrigodamottacc}@gmail.com  [2]jesuino.vieira@umontreal.ca

All authors contributed equally

## Abstract

Semantic representations can be framed as a structured, dynamic knowledge space through which humans navigate to retrieve and manipulate meaning. To investigate how humans traverse this geometry, we introduce a framework that represents concept production as navigation through embedding space. Using different transformer text embedding models, we construct participant-specific semantic trajectories based on cumulative embeddings and extract geometric and dynamical metrics, including distance to next, distance to centroid, entropy, velocity, and acceleration. These measures capture both scalar and directional aspects of semantic navigation, providing a computationally grounded view of semantic representation search as movement in a geometric space. We evaluate the framework on four datasets across different languages, spanning different property generation tasks: Neurodegenerative, Swear verbal fluency, Property listing task in Italian, and in German. Across these contexts, our approach distinguishes between clinical groups and concept types, offering a mathematical framework that requires minimal human intervention compared to typical labor-intensive linguistic preprocessing methods. Comparison with a non-cumulative approach reveals that cumulative embeddings work best for longer trajectories, whereas shorter ones may provide too little context, favoring the non-cumulative alternative. Critically, different embedding models yielded similar results, highlighting similarities between different learned representations despite different training pipelines. By framing semantic navigation as a structured trajectory through embedding space, bridging cognitive modeling with learned representation, thereby establishing a pipeline for quantifying semantic representation dynamics with applications in clinical research, cross-linguistic analysis, and the assessment of artificial cognition. https://github.com/jesuinovieira/semtraj-iclr2026

## 1 Introduction

Semantic representations are the stored, structured traces of our knowledge about the world (Hills et al., 2015). Retrieving a concept depends on context and draws jointly on experiential details and abstract, shared knowledge, for "dog," this might range from memories of a family pet to generic category knowledge (Xie et al., 2024; Barsalou, 2023). Navigation in semantic representations involves searching within a space that is both dynamic and context-dependent, including features for sensorimotor representations, affective experiences, linguistic encoding, and contextual cues (Hills et al., 2015; Diveica et al., 2025).

Influential cognitive models of semantic navigation treat this search as a foraging process, where humans balance exploitation and exploration (Hills et al., 2012), much like animals foraging for resources. Operationally, this dynamic has been measured using two primary components of semantic control: clustering and switching (Troyer et al., 1997). Clustering (exploitation) refers to the generation of items within a specific semantic subcategory (e.g., 'cat,' 'dog' within 'house animals'),

while switching (exploration) marks the transition to a new subcategory (e.g., moving from 'house animals' to 'sea animals'). Crucially, this navigation is an inherently cumulative process. Successful semantic retrieval relies on executive functions, specifically working memory and inhibitory control, to monitor the search history and inhibit the repetition of previously generated items (Baddeley, 2000; Unsworth et al., 2011). Consequently, the semantic representation of any given response is not isolated; rather, it is contextually bound to the sequence of preceding words.

Here, we extend this approach and adopt the view that semantic retrieval can be understood as navigation through a multidimensional space in which multiple features jointly define each concept. From this perspective, we move beyond the binary distinction of clustering and switching. Instead, we aim to capture the step-by-step granularity of the search process itself by computing navigation metrics. To capture this history-dependent dynamic, we propose a natural-language–based characterization of human semantic navigation using cumulative embeddings, modeling the search as trajectories in transformer-based embedding space.

Classical task paradigms in cognitive sciences such as semantic fluency and property listing provide behavioral windows into this search process (Canessa et al., 2024; Canessa & Chaigneau, 2020; Troyer et al., 1997), and formal models have described how people balance exploitation and exploration over time (Hills et al., 2012). Yet these approaches often rely on labor-intensive, heterogeneous pipelines that hinder comparability across studies (Chaigneau et al., 2018). Natural Language Processing (NLP) methods—especially embedding-based analyses—offer scalable alternatives that have already helped differentiate clinical groups and organize conceptual structure (García et al., 2025); for example, word embedding metrics have separated Alzheimer's and Parkinson's patients from controls, and language model (LM) embedding trajectories have characterized psychosis and schizophrenia profiles (Toro-Hernández et al., 2024; Ferrante et al., 2025; He et al., 2024; Nour et al., 2023; Lopes da Cunha et al., 2024; Sanz et al., 2022; Palominos et al., 2024).

Building on this foundation, our framework represents concept production as movement through transformer-based spaces. For each participant, we construct semantic trajectories and extract geometric and dynamical metrics, distance to next, velocity, acceleration, entropy, distance to centroid, that capture both scalar and directional aspects of navigation. Critically, this trajectory-based approach offers a continuous view of the search, complementing traditional metrics of semantic control by capturing a more fine-grained navigation dynamics. Furthermore, this computational approach minimizes manual intervention while preserving rich structure in the data, enabling principled tests of hypotheses about semantic meaning and search in humans and in artificial agents (Xu et al., 2025).

We demonstrate the effectiveness of our approach by applying it to datasets that specifically challenge standard LM embeddings. Our evaluation probes: the clinical utility of embeddings for analyzing natural language in patients with Parkinson's disease (Linz et al., 2017); the semantic consistency of multilingual embeddings across Italian and German (Conneau et al., 2020; Artetxe & Schwenk, 2019); and the atypical geometric properties of swear word embeddings as revealed through a verbal fluency task (Graumas et al., 2019). Critically, our metrics provide novel insights in each of these established research areas by isolating the specific trajectory features that differentiate between different groups and semantic categories. Notably, different embedding models yield essentially similar patterns, suggesting convergent geometry across learned representations despite distinct training pipelines and architectural differences (Valeriani et al., 2023; Doimo et al., 2024; Lee et al., 2025; Wolfram & Schein, 2025). By framing human semantic retrieval as structured trajectories in embedding space, we bridge cognitive modeling with learned representations and establish a pipeline for quantifying semantic dynamics with applications to clinical research and cross-linguistic analysis (Shakeri & Farmanbar, 2023). This approach holds promise for applications, including the classification of brain disorders, the differentiation between concept types, and the testing of core hypotheses about search dynamics in artificial agents, as models that compare human responses to linguistic data with LLMs' generated responses.

## 2 METHODS

### 2.1 DATASETS

To evaluate our metrics, we use four open datasets that vary in language, population, and tasks. Table 1 provides an overview of their key characteristics and descriptive statistics.

Table 1: Overview of the four datasets used in our experiments. We report the mean $\pm$ standard deviation for the number of properties and words produced per concept–participant pair.

| Dataset | Language | Subjects | Comparisons | Properties | Words |
|---|---|---|---|---|---|
| Neurodegenerative | ES | 76 | 3 | $19.53 \pm 12.39$ | $19.79 \pm 12.64$ |
| Swear Fluency | EN | 274 | 5 | $20.69 \pm 7.88$ | $21.20 \pm 8.10$ |
| Italian | IT | 69 | 10 | $4.96 \pm 1.86$ | $14.14 \pm 6.81$ |
| German | DE | 73 | 10 | $5.49 \pm 1.82$ | $14.31 \pm 6.00$ |

**Neurodegenerative dataset.** Introduced in Toro-Hernández et al. (2024), consists of 76 Chilean Spanish-speaking participants divided into three groups: individuals with Parkinson's disease (PD), with the behavioral variant of frontotemporal dementia (bvFTD), and healthy controls (HC). Participants completed a property listing task (PLT), in which they were asked to generate as many attributes as possible for 10 concrete concepts ("tree," "sun," "clown," "puma," "airplane," "hair," "duck," "house," "shark," and "bed"). Instructions emphasized the inclusion of "physical characteristics, internal parts, appearance, sounds, smells, textures, uses, functions, and typical locations" (Toro-Hernández et al., 2024). Finally, the data in this paper was preprocessed by only extracting content words (nouns, verbs, adjectives, and adverbs).

**Swear fluency dataset.** Introduced by Reiman & Earleywine (2023), includes 274 undergraduate native speakers of U.S. English who performed verbal fluency tasks across domains. In this case, participants were instructed to generate as many items as possible within a given category in one minute (e.g., if the category was "animals," acceptable responses might include "dog," "cat," "lion," or "tiger"). The categories comprised animals, words beginning with F, A, and S, and swear words.

**Italian and German datasets.** Drawn from Kremer & Baroni (2011), comprising 69 Italian and 73 German students, respectively. Participants were asked to generate descriptive properties for 50 concrete concepts, divided into 10 categories: : "Bird," "Body Part," "Building," "Clothing," "Fruit," "Furniture," "Implement," "Mammal," "Vegetable," and "Vehicle". The task has a time limit of one minute per item. Participants were encouraged to provide at least four descriptive phrases per concept and were not allowed to return to previously described items once the time expired.

## 2.2 CHARACTERIZING NAVIGATION

Participants generate concept streams—ordered lists of items (e.g., "cat," "dog," ...)—of length $N$. Let item $t$ denote the $t$-th entry. We map each stream to a trajectory in semantic space, $X = (x_1, \ldots, x_N)$, where $x_t$ is the point associated with item $t$. The points are time-indexed ($x_1$ is the first item, $x_2$ the second, etc.). Rather than computing embeddings independently (Linz et al., 2017; Nour et al., 2023), we construct them cumulatively: $x_t$ summarizes items 1:$t$. For example, if the first two items are "cat" then "dog," $x_2$ encodes "cat dog." This design captures dependencies among successive items, avoids independence assumptions, and yields a distinct trajectory for each participant–concept pair, enabling analysis of navigation dynamics (Figure 1). Because $x_t$ conditions on the full prefix, this approach calls for more complex, sequence aware embedding representations capable of modeling memory dependent semantics.

**Distance to Next.** To quantify moment-to-moment change in semantic state, we compute the cosine distance between each pair of successive unit-normalized embeddings, yielding an $N-1$ length series of step sizes ("semantic jumps") per trajectory. Larger values indicate bigger shifts in meaning from one item to the next. Because trajectories naturally differ in length, we also summarize each series with its mean step size, the average cosine distance across step, as a length-invariant indicator of average memory-search breadth. In cognitive terms, this metric assesses the local, step-by-step dynamic of the search. For example, the 'semantic jump' from 'dog' to 'shark' would yield a larger distance than the jump from 'cat' to 'dog'.

**Entropy.** We also summarize the information contained of the distance to next with a scale-free approximate Shannon entropy. Distances are split at their within-trajectory median into "high" versus "low," forming a binary sequence whose Shannon entropy is computed and then normalized by the number of valid steps (Pincus et al., 1991). This value is set to zero when all steps fall

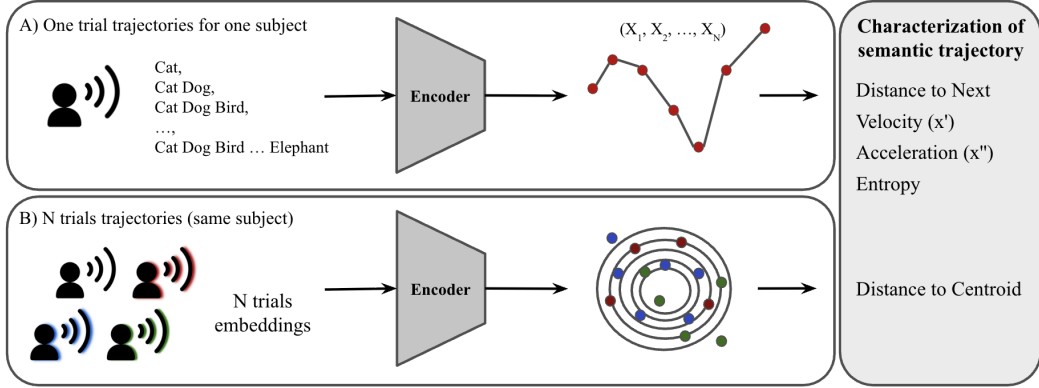

Figure 1: A schematic of the semantic trajectory analysis. (A) In a single trial, a participant generates a cumulative word list. A text encoder then maps each sequential step to a vector embedding, creating a trajectory in semantic space. This path is characterized using dynamical metrics like velocity ($x'$), acceleration ($x''$), and entropy. (B) Across multiple trials for the same subject, the dispersion of the resulting cloud of embeddings trajectories is summarized by measuring the distance of each point to the collective centroid.

on a single side of the median and is only estimated when at least three valid steps are available, ensuring stability for short sequences. Let $\theta$ denote distance to next within-trajectory median. We first binarize the sequence:

$$b_t = \begin{cases} 1, & x_t \geq \theta, \\ 0, & x_t < \theta, \end{cases} \qquad t = 1, \ldots, n. \tag{1}$$

Let $p = \frac{1}{n} \sum_{t=1}^{n} b_t$ be the fraction of ones. The Shannon entropy of the binarized sequence is

$$H = -p \log_2 p - (1-p) \log_2(1-p), \tag{2}$$

with the convention $H = 0$ when $p \in \{0, 1\}$. This measure can be interpreted as the information richness of fluctuation around a typical step size in a time series (Gao et al., 2008). Theoretically, high entropy indicates a less predictable, more variable search dynamic, that may be consistent with higher cognitive effort or reduced executive control. This suggests a disorganized or erratic search process (e.g., the sequence 'cat', 'shark', 'dog' would be more entropic than 'cat', 'dog', 'shark'), potentially reflecting a failure to maintain a consistent search strategy.

**Velocity and Acceleration.** Beyond scalar cosine distances, we characterize the semantic directional dynamics by computing discrete derivatives of the embeddings themselves, similarly to Nour et al. (2025). It is important to remark that we are assuming a Euclidean dynamics for simplicity, which overlooks the real anisotropic nature of the embedding spaces (Nickel & Kiela, 2017; Ethayarajh, 2019). Velocity is defined as the vector difference between consecutive embeddings, yielding both a direction and a magnitude for each step; the final row has no velocity. Since in the datasets tasks don't have a time stamp, then $\alpha = \Delta t^{-1} = 1$.

$$\mathbf{v}_t = \alpha(\mathbf{x}_{t+1} - \mathbf{x}_t), \quad (t = 1, \ldots, T-1) \tag{3}$$

Acceleration is defined as the difference between successive velocity vectors and quantifies changes in direction or speed from one step to the next; the final two rows have no acceleration. These kinematic quantities retain information about where the trajectory is heading in the high-dimensional space—information that step-wise scalar distances alone cannot convey. By default, derivatives assume a unit time step between items; if timestamps are available, magnitudes can be rescaled accordingly with $\alpha$.

$$\mathbf{a}_t = \alpha(\mathbf{v}_{t+1} - \mathbf{v}_t) = \alpha^2(\mathbf{x}_{t+2} - 2\mathbf{x}_{t+1} + \mathbf{x}_t), \quad (t = 1, \ldots, T-2) \tag{4}$$

Crucially, velocity quantifies the magnitude and direction of a single step (e.g., 'dog' to 'shark' yields high velocity). Acceleration, in turn, quantifies the change in this movement. In cognitive terms, stable 'exploitation' (clustering) involves minimal changes in velocity, thus yielding low acceleration.

Conversely, a process characterized by 'unstable exploitation' (a series of erratic 'switches') would involve constant, significant changes in magnitude, thus resulting in a high average acceleration as a physics inspired characterization.

**Distance to Centroid.** To capture how individual properties relate to the overall semantic context, we computed a centroid-based measure. When categorical property labels and their embeddings were available, repeated occurrences of the same property were collapsed to a single instance, ensuring that redundancy did not overweight specific properties. For each unique property, we retained only its first embedding and constructed a centroid vector representing the average position of all unique property embeddings over N trials in each specific concept and unique subject. Each item in the sequence was then assigned the cosine distance between its embedding and this centroid. This measure quantifies how far each produced property lies from the central tendency of the participant's semantic exploration, providing an index of dispersion that complements step-wise trajectory distances. A high average distance to centroid indicates a more dispersed search, suggesting that the participant explored a wider, more varied semantic space. Conversely, a low distance to centroid would suggest a more focused or 'centered' search, confined to a tighter semantic neighborhood.

In sum, these five metrics are complementary, each characterizing a distinct aspect of the navigation process inspired by physics. Together, they provide a granular view of a person's semantic organization and executive control. Specifically, Distance to Next, Velocity, and Acceleration quantify the local kinematics of the search; Entropy quantifies the global variability and predictability of the trajectory; and Distance to Centroid captures the global dispersion of the entire search relative to a central meaning. This multi-metric approach allows us to answer how the search unfolds.

### 2.2.1 BASELINE AND EMBEDDING MODEL COMPARISON

Each trajectory is a time-ordered sequence of dense multilingual embeddings. Unless otherwise noted, all results are reported using OpenAI's text-embedding-3-large; results with alternative encoders (Google's text-embedding-004 and Qwen3-Embedding-0.6B, and using fastText as baseline). To benchmark our approach against traditional static representations, we computed all metrics using non-cumulative fastText embeddings, serving as a baseline for standard semantic organization models (Mikolov et al., 2018). Moreover, the model choice uses causal attention, in case of Qwen3 (Zhang et al., 2025) and bidirectional-encoder Gecko in Google's text-embedding-004 Lee et al. (2024); however, OpenAI's model doesn't have any documentation on the training method. Finally, to address the potential confound of embedding anisotropy, we applied a ZCA-whitening transformation to our embeddings and re-calculated the metrics to assess the robustness of group discrimination under approximated isotropic conditions (Bell & Sejnowski, 1997; Zhuo et al., 2023; Su et al., 2021). These results are included in the Appendix A.4 due to only minor differences between groups. Moreover, Qwen3, Google's text-embedding-004 and fastText models are also reported in the Appendix A.

### 2.2.2 STATISTICAL ANALYSIS

For each metric, we evaluated group- and concept-level effects using generalized linear mixed models (GLMMs), with each metric as a fixed factor, including participants and concept as random factors to account for repeated measures and individual variability. Models were fitted according to the most appropriate distribution. Based on this procedure, a lognormal distribution was applied to distance to next, entropy, velocity, and acceleration, while a Gaussian distribution was used for the distance-to-centroid metric. Post-hoc pairwise comparisons were adjusted with Tukey's HSD to control the family-wise error rate. Visualizations combine raw distributions (boxplots with jittered points) with model-estimated marginal means and $95\%$ confidence intervals, annotated with significance levels from the Tukey tests. This approach highlights both the variability in the data and the inferential estimates used for statistical testing. All statistical analyses were conducted in R (v.4.3.1) using the glmmTMB package (Brooks et al., 2017).

# 3 RESULTS

## 3.1 NEURODEGENERATIVE DATASET

There was a significant effect of category across all metrics. For distance to next, healthy controls showed lower values than both bvFTD and PD, while bvFTD and PD did not differ. A similar pattern emerged for velocity and for acceleration: HCs were lower than both patient groups, with bvFTD and PD comparable. For entropy, HC again showed reduced values compared to both groups, with no difference between bvFTD and PD. In contrast, distance-to-centroid showed the opposite pattern: HC exhibited greater distances than both patient groups, which did not differ from each other. These results indicate that semantic navigation in patient groups is characterized by greater spread, higher variability, increased entropy, and more compact clustering relative to controls (see Figure 2).

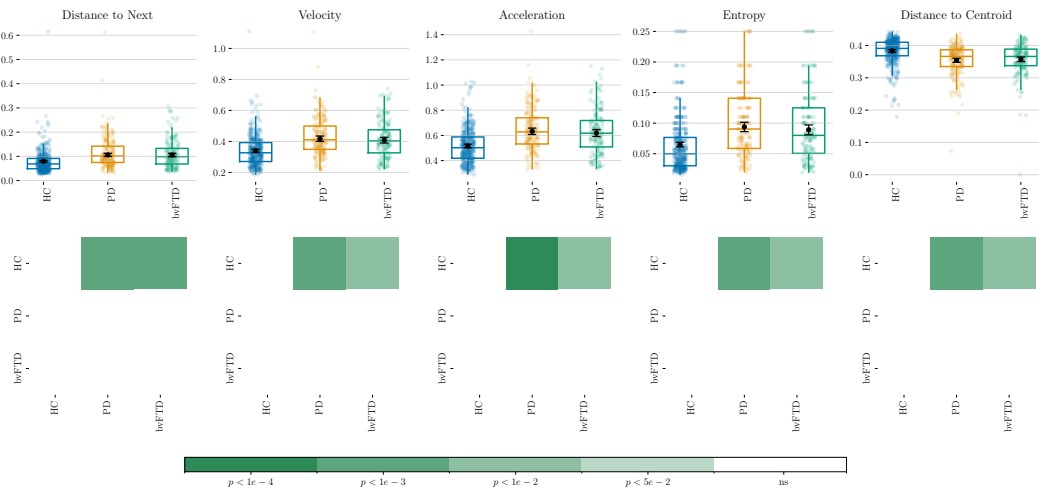

Figure 2: Boxplot of Neurodegenerative metrics by category. Matrices display pairwise statistical comparisons.

## 3.2 SWEAR FLUENCY DATASET

For distance to next, all letter categories and swear words were higher than animals, with swear words eliciting the longest distance to next and animals the shortest; letters occupied an intermediate range. Velocity showed the same pattern, with navigation being faster for letters and especially for swear words than for animals. Acceleration mirrored velocity, with letters and, most prominently, swear words exceeding animals. For entropy, animals showed the lowest values; letters were higher, and swear words the highest. Finally, distance-to-centroid reversed the pattern, with animals being farther from the centroid than letters, whereas swear words were markedly closer. Overall, swear words consistently drove the strongest responses across metrics, animals the lowest, and letters clustered in between (see Figure 3).

## 3.3 ITALIAN DATASET

Relative to Bird (reference group), most categories showed shorter distance to next, with Building and Vehicle the least separated from Bird. Velocity followed the same ordering, with Bird exceeding most categories and Building and Vehicle only weakly or not separated. Acceleration mirrored velocity, again showing Bird higher than the bulk of categories. Entropy differences were selective: several categories had lower entropy than Bird, whereas many contrasts were not significant. Distance-to-centroid showed a partially reversed structure: some categories were farther from the centroid than Bird, while others (e.g., Body Part, Clothing, Implement) were closer; several categories showed no difference from Bird (see Figure 4).

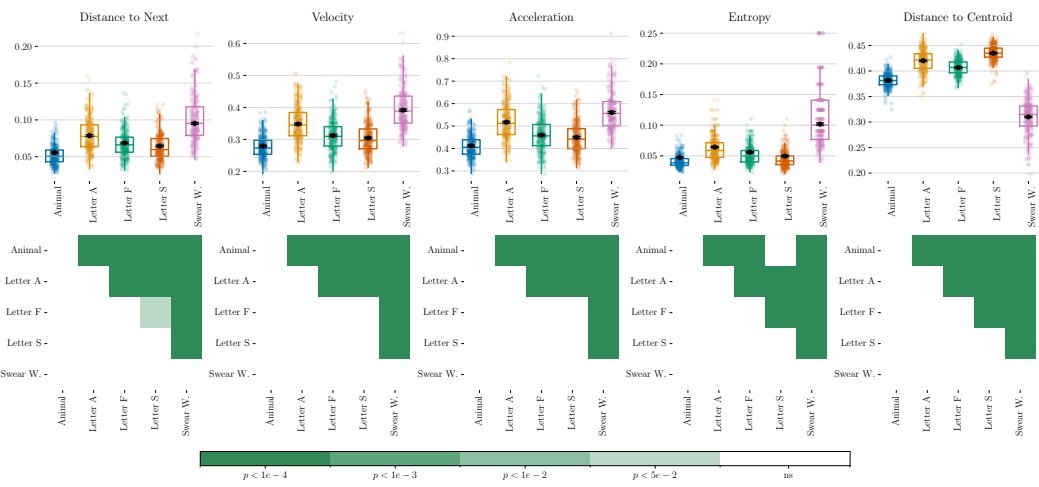

Figure 3: Boxplot of Swear metrics by category. Matrices display pairwise statistical comparisons.

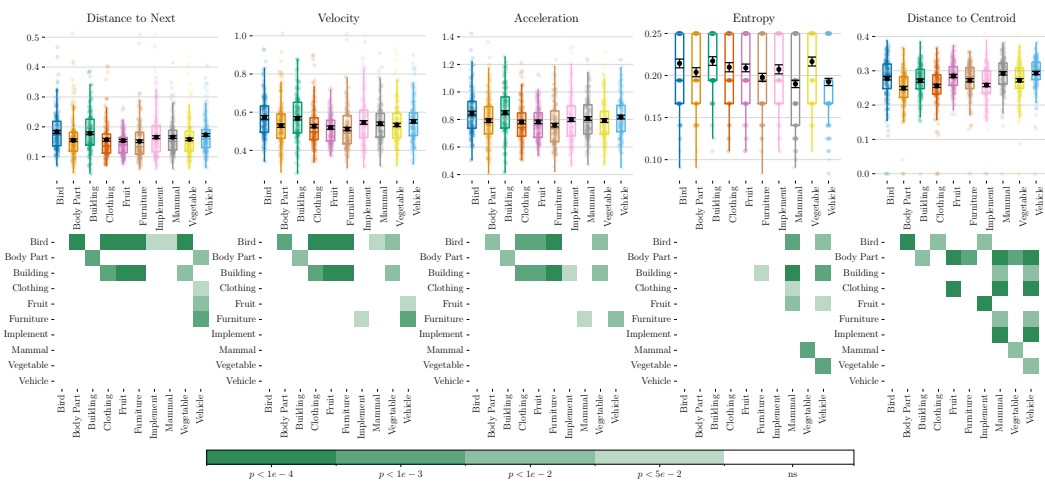

Figure 4: Boxplot of Italian metrics by category. Matrices display pairwise statistical comparisons.

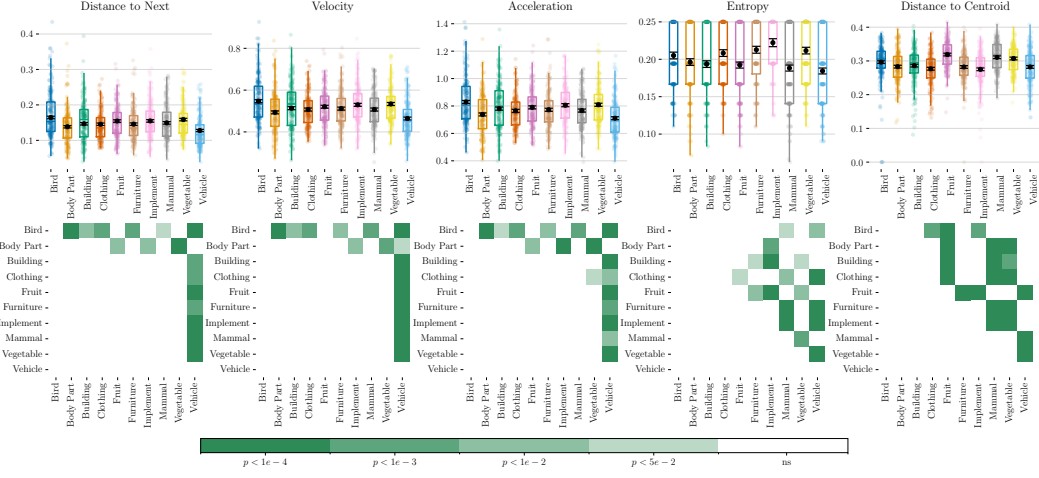

Figure 5: Boxplot of German metrics by category. Matrices display pairwise statistical comparisons.

## 3.4 German dataset

For distance next, most categories produced shorter values than Bird, with Vehicle and several others most distinct from Bird, and Vegetable showing little to no separation. Velocity showed Bird higher than nearly all categories, with the largest gap against Vehicle. Acceleration followed the same pattern, with Bird exceeding most categories and the clearest separation against Vehicle. Entropy differences were selective: several categories were lower than Bird, while Implement was higher; many others showed no differences. Finally, distance-to-centroid revealed a different structure: some categories (e.g., Fruit, Mammal, Vegetable) were farther from the centroid than Bird, whereas others (e.g., Body Part, Building, Clothing, Furniture, Implement, Vehicle) were closer, with the most pronounced gaps involving Fruit compared to clothing and tool-like categories (see Figure 5).

## 3.5 Model Comparison

To ensure our findings were not dependent on a specific text encoder, we compared the trajectory metrics from three different models. The non-cumulative fastText baseline is reported in Appendix A.5, and cumulative vs. non-cumulative comparisons for all models are provided in Table 2. Figure 6 shows the Pearson correlation matrices for five metrics across all four datasets, revealing a generally high degree of robustness. The results exhibit a block-diagonal structure, indicating strong positive correlations for each metric across models. Kinematic measures—velocity, acceleration, and distance-to-next—are highly correlated across models and moderately correlated with one another, consistent with their shared capture of step-wise trajectory dynamics (except the baseline; fastText is its non-cumulative version). Two metrics diverge: distance to centroid and entropy. The first shows the weakest inter-model correlation, we attribute the low inter-model correlation of distance-to-centroid to its sensitivity to embedding geometry. Phenomena such as anisotropy (Ethayarajh, 2019) and 'rogue dimensions' (Timkey & van Schijndel, 2021) result in unique, model-specific centroids that distort global statistics. Whereas entropy shows near-perfect inter-model correlation because it depends on rank ordering rather than absolute distances; median binarization further stabilizes it when models agree on the relative size of semantic jumps.

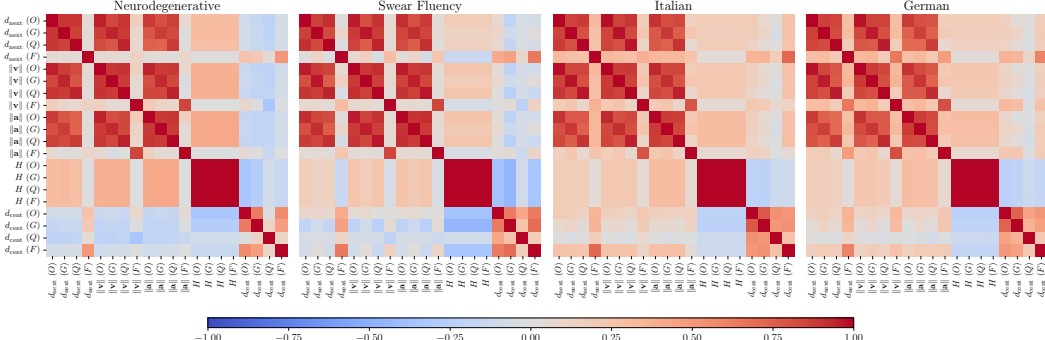

Figure 6: Pearson correlations of metrics across datasets. From left to right: Neurodegenerative, Swear Fluency, Italian, and German. Each heatmap reports pairwise correlations between Distance to Next, Velocity, Acceleration, Entropy, and Distance to Centroid, computed with three embedding models (O = OpenAI text-embedding-3-large, G = Google text-embedding-004, Q = Qwen3-Embedding-0.6B, F = fastText). Color indicates correlation strength (blue = negative, red = positive).

## 4 Discussion

We aimed to characterize human semantic navigation through a fine-grained analysis of step-by-step dynamics. Our results indicate that this approach captures granular aspects of the search process that are complementary to traditional clustering and switching dynamics (Troyer et al., 1997). By successfully differentiating between clinical groups and concepts, our framework offers a highly scalable, automated technique. This addresses critical challenges in the field regarding automation, reliability, and the limitations of time-consuming manual pipelines (Buchanan et al., 2020; Ramos

et al., 2024; 2025). Crucially, our work builds upon previous literature that successfully leveraged embedding spaces for semantic characterization (Linz et al., 2017; Toro-Hernández et al., 2024; Sanz et al., 2022; Zhang et al., 2022). We further extend these approaches by explicitly modeling the search trajectory, using step-by-step geometric and kinematic metrics, inspired by physics, as a dynamic proxy for human semantic navigation. Our cumulative transformer-based characterization approach enhances prior NLP-based methods for human semantic navigation (Linz et al., 2017), particularly for tasks involving large sequences, as shown in A.3.

To test our approach, we tested 4 datasets. First, the analysis of the neurodegenerative dataset reveals a semantic navigation profile primarily characterized by disorganization, reflecting the executive dysfunction typical of these clinical groups. While healthy navigation relies on strategic control to balance clustering and switching, patients with PD and bvFTD often suffer from executive constraints that disrupt this regulation (Birba et al., 2017; Cousins & Grossman, 2017). Our metrics capture this deficit as a specific kinematic signature: rather than focused exploitation, patients exhibited significantly higher distance to next, velocity, and acceleration. These elevated values indicate a volatile trajectory, marked by erratic 'semantic jumps' and abrupt changes in direction, consistent with an inability to sustain stable semantic clusters, or effectively inhibit irrelevant associations. Crucially, this disordered traversal is further confirmed by higher entropy, quantifying a search process that is significantly less predictable and routine than that of controls, thereby extending the semantic variability findings of the original work (Toro-Hernández et al., 2024). Finally, distance to centroid results showed that, despite their kinematic volatility, patient searches were spatially more constricted, remaining closer to a central meaning. This likely reflects a diminished semantic space, prompted by the loss of sensorimotor traces required to evoke rich content (Cousins & Grossman, 2017; Fernandino et al., 2013; Boulenger et al., 2008), which, combined with the aforementioned executive dysfunction, restricts navigation to a narrower semantic neighborhood (Zhang et al., 2022).

Conversely, the swear fluency dataset revealed a distinct navigational profile characterized by a highly variable semantic exploration within a constrained space. Navigation for curse words was marked by significantly larger 'semantic jumps' among successive items, reflected in higher velocity and acceleration dynamics. This kinematic volatility, combined with higher entropy, aligns with the unique organization of taboo lexicons: unlike taxonomic categories (e.g., animals) which allow for structured exploitation of sub-categories, the structure of swear words -and differential search strategies- may lead to more erratic retrieval (Jay & Jay, 2015). Regarding global geometry, swear-word fluency exhibited a lower distance to centroid than animal and letter tasks. This higher centrality is consistent with the structural nature of the taboo category, a lexical neighborhood that is spatially compact but explored with high variability and directional unpredictability.

Furthermore, the analysis of the Italian and German datasets revealed that kinematic patterns consistently exposed an informative structure regarding how retrieval unfolds within specific linguistic contexts across models. Since data acquisition followed an identical protocol for both languages (Kremer & Baroni, 2011), observed differences cannot be ascribed to task administration artifacts. While our results successfully discriminated specific semantic categories, differences in category effects emerged between the Italian and German datasets. As noted in psycholinguistic research, variables such as concreteness and specificity are encoded differently depending on linguistic structure (Bolognesi et al., 2020; Montefinese et al., 2023). This reflects the flexible nature of lexico-semantic representations, where cultural conventions shape how meaning is accessed and organized during semantic search Vigliocco et al. (2009); Barsalou (2023); Kemmerer (2023). Thus, different transformer-based models, trained on distinct corpora, may be expected to capture specific manifestations of semantic structure in divergent ways (Conneau et al., 2020; Artetxe & Schwenk, 2019). Our results open new avenues in less-explored domains as a general approach, such as the semantic navigation underlying the production of swear words. Since swear-word fluency has been linked to substance use (Reiman & Earleywine, 2023) and differential brain activity patterns in schizophrenia (Lee et al., 2019), analyzing its semantic dynamics could provide novel insights into behavioral regulation and inhibitory control.

Crucially, our cross-model analyses revealed that trajectory metrics were highly correlated across the three different embedding models, despite their training differences (i.e., causal and bidirectional-encoder attention), indicating that the observed dynamics are not an artifact of a single model, as agrees with previous literature as they generate similar representations (Lee et al., 2025; Wolfram & Schein, 2025). This consistency was particularly strong for metrics capturing the local, step-by-step

evolution of the trajectory, such as velocity, acceleration, and entropy. In contrast, the distance-to-centroid metric consistently showed the lowest inter-model correlation, revealing that while models agree on a trajectory's local dynamics (its shape and variability), they differ significantly in its global positioning. This sensitivity arises because the metric uses a static, global average rather than successive states, making it dependent on the unique high-level geometry of each model's embedding space. It might be a possible tool for comparing how different models structure knowledge. Notably, this dissimilarity between models was most pronounced in the neurodegenerative dataset, potentially reflecting a more complex disruption in semantic navigation.

## 5 Conclusion

In sum, applying five cumulative embedding-based trajectory metrics, inspired by physics, to fluency and property listing tasks data are complementary to previous NLP methods, reveling signatures of semantic navigation: distance to next, velocity, and acceleration distinguished neurodegenerative groups from healthy controls in their semantic search; entropy captured irregularity in search (notably higher for swear-word fluency); and distance to centroid indexed positional centrality orthogonal to the other measures, exposing category and language-specific structure. These effects were broadly consistent across three multilingual, causal and bidirectional, transformer embedding models, indicating robustness to the choice of encoder for local trajectory dynamics, while lower cross-model agreement for distance-to-centroid highlights model-dependent global geometry. Together, these results show a geometrically grounded NLP framework for characterizing human semantic retrieval, through across tasks and languages, and open avenues for clinical stratification and cross-model comparisons of how humans and LMs navigate semantic space.

## 6 Limitations and Future Work

Although fluency and property-listing tasks are useful across a range of applications, they capture only a partial view of human semantic navigation, in this specific case the tasks didn't contain the time step of the words. This could contribute to more temporal meaningful dynamics. Developing richer speech-based protocols may help probe semantic search and representation more directly, especially when linked to learned representations from language models (LMs). We acknowledge that our assumption of Euclidean dynamics is a simplification that overlooks the anisotropic structure of embedding spaces (Nickel & Kiela, 2017; Ethayarajh, 2019) and the tests with ZCA-Whitening showed only little difference between groups, indicating a low effect of the anisotropy A.4. More mathematically robust, potentially non-Euclidean, metrics can be used to better characterize these trajectories. Furthermore, we used a basic information measure (Shannon entropy); future work should broaden the information-theoretic toolkit for semantic navigation to assess complexity in systems with many interacting variables that jointly shape human semantic representation and retrieval. Finally, future work could apply a similar framework to characterize different LLMs and assess their generative semantic navigation across tasks. The goal is to develop a unified account of trajectories in semantic space that encompasses both humans and generative language models.

## Reproducibility Statement

The code and data required to reproduce the findings of this study are openly available. The source code for all analyses and figure generation is accessible at https://github.com/jesuinovieira/semtraj-iclr2026. The datasets are all from public sources, which are detailed with their respective access links in Appendix A.2.

## Acknowledgments

Felipe Diego Toro-Hernández acknowledges financial support from the São Paulo Research Foundation (FAPESP), grant 2025/04243-2.

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

# A  APPENDIX

Figures in Appendices A.5, A.6, and A.7 share a common format: boxplots with jittered observations and a Tukey HSD pairwise comparison matrix. To maintain clarity without overloading the appendix with redundant details, captions specify only the dataset and embedding model.

## A.1  USE OF LARGE LANGUAGE MODELS DISCLOSURE

We used large language models (LLMs) as a code assistant and text editor to refine implementation details, improve manuscript clarity and grammar, and identify relevant literature. All LLM outputs were thoroughly reviewed and verified by the authors. The conceptual framework and methodology contributions presented in this work are entirely our own.

## A.2  DATA AVAILABILITY

The datasets used in this study are publicly available. The neurodegenerative dataset can be found at `https://osf.io/8pufk/`, and the swear words dataset is located at `https://osf.io/w8drt/`. The Italian and German datasets were sourced from the appendices of Kremer & Baroni (2011), available at `https://link.springer.com/article/10.3758/s13428-010-0028-x`.

## A.3  NON-CUMULATIVE EMBEDDING RESULTS

Table 2 compares the cumulative and non-cumulative versions of each embedding model across the four datasets. Overall, we observe a consistent pattern that aligns with the structure and length of the underlying trajectories.

For the Neurodegenerative dataset, all models clearly benefit from the cumulative representation: the cumulative variants yield more significant pairwise differences and higher effect sizes. This suggests that longer trajectories allow cumulative embeddings to capture broader semantic patterns that are not as evident when looking only at point-to-point changes. The Swear Fluency dataset shows a milder version of this effect. The cumulative models still identify more significant differences, but the non-cumulative variants produce slightly larger Cohen's $d$.

In contrast, the Italian and German datasets favor the non-cumulative variants, which achieve the highest number of significant differences and the strongest effect sizes across models. This pattern is consistent with the much shorter trajectories in these datasets (Table 1). When sequences contain very few observations, there is less contextual information to average over, and the geometric metrics

Table 2: Summary of significant differences across embedding backends. For each dataset, we report the number of significant difference pairs (# Pairs; binary count, with the weighted count—ranging from 1 to 4 depending on the significance level—in parentheses) and the mean Cohen's $d$ across metrics. Rows marked with (nc) indicate non-cumulative variants. Bold values indicate the highest result in each column.

| Model | Neurodegenerative | | Swear Fluency | | Italian | | German | |
|---|---|---|---|---|---|---|---|---|
| | # Pairs | $d$ | # Pairs | $d$ | # Pairs | $d$ | # Pairs | $d$ |
| openai 003 | **10** (27) | **0.30** | **46** (185) | 0.59 | 63 (203) | 0.18 | 80 (284) | 0.21 |
| google 004 | 7 (21) | 0.28 | 45 (182) | 0.58 | 81 (306) | 0.22 | 110 (413) | 0.28 |
| qwen | 9 (25) | 0.26 | 43 (162) | 0.40 | 84 (304) | 0.22 | 112 (413) | 0.24 |
| fasttext | 8 (22) | 0.23 | 46 (182) | 0.72 | 81 (311) | 0.22 | 105 (397) | 0.23 |
| openai 003 (nc) | 4 (10) | 0.19 | 46 (182) | 0.64 | 81 (281) | 0.24 | 111 (413) | 0.28 |
| google 004 (nc) | 4 (13) | 0.19 | 40 (161) | 0.76 | **107** (395) | **0.29** | **137** (521) | **0.37** |
| qwen (nc) | 4 (11) | 0.15 | 40 (148) | 0.39 | 92 (353) | 0.28 | 114 (427) | 0.27 |
| fasttext (nc) | 4 (10) | 0.17 | 37 (143) | **0.86** | 70 (277) | 0.25 | 122 (471) | 0.31 |

naturally exhibit greater point-to-point variation. Non-cumulative embeddings retain this variability, which helps them expose stronger cross-model differences.

Taken together, these results suggest that the choice between cumulative and non-cumulative embeddings depends on trajectory length: longer sequences benefit from cumulative aggregation, whereas shorter sequences retain more discriminative signal under a non-cumulative representation.

## A.4 RESULTS IN ZCA-WHITENED EMBEDDING SPACE

We conducted a preliminary analysis using the OpenAI embedding model to assess the impact of correcting for anisotropic geometry via ZCA-whitening (Bell & Sejnowski, 1997; Zhuo et al., 2023; Su et al., 2021). Before computing metrics, we verified the transformation by checking the covariance matrix: diagonals were near 1 and off-diagonals near 0. For the Parkinson's and Swear Fluency datasets, the metrics derived from whitened embeddings were nearly identical to those obtained from the original cumulative embeddings. In contrast, the Italian and German datasets exhibited slight, non-directional fluctuations. This variation is likely explained by the significant difference in trajectory lengths across datasets. As shown in Table 1, the mean number of properties per trajectory is relatively high for the Parkinson's (19.53) and Swear Fluency (20.69) datasets, whereas the Italian and German datasets feature much shorter trajectories (4.96 and 5.49, respectively). These shorter sequences inherently yield higher metric variability, rendering them less stable than the robust estimates derived from longer trajectories.

Figure 7 summarizes the impact of whitening on the correlation structure. ZCA introduces only minimal changes, indicating that the relative relationships between metrics remain stable despite the transformation. This supports the robustness of our semantic trajectory analysis under both raw and whitened embedding spaces, though further investigation is needed to determine when explicit anisotropy correction is beneficial.

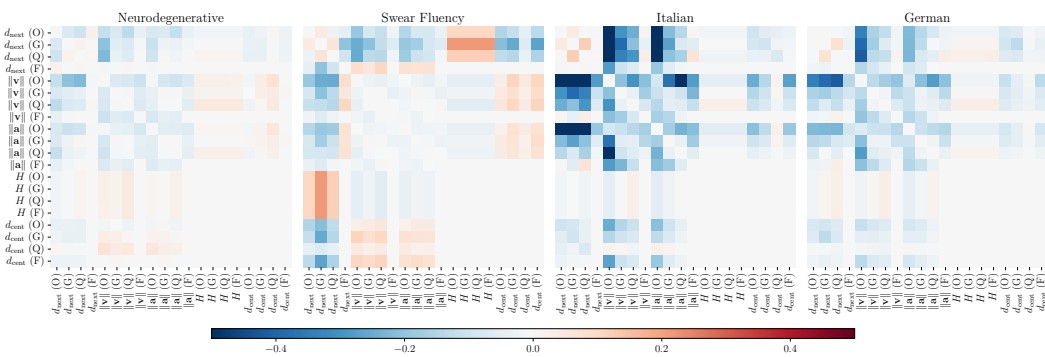

Figure 7: Impact of ZCA Whitening on metric correlation structure. Heatmaps showing the difference in Pearson correlations (ZCA - Raw) between semantic trajectory metrics computed from ZCA-whitened versus raw embeddings across four datasets (Neurodegenerative, Swear Fluency, Italian, German) and four embedding models (O = OpenAI text-embedding-3-large, G = Google text-embedding-004, Q = Qwen3-Embedding-0.6B, F = fastText). Each cell represents the change in correlation between two metric-model pairs, where red indicates increased correlation after whitening and blue indicates decreased correlation.

## A.5 FASTTEXT RESULTS (BASELINE)

All metrics here are computed using the non-cumulative fastText embeddings, which we adopt as our baseline. For each dataset, we employ the corresponding language-specific fastText model.

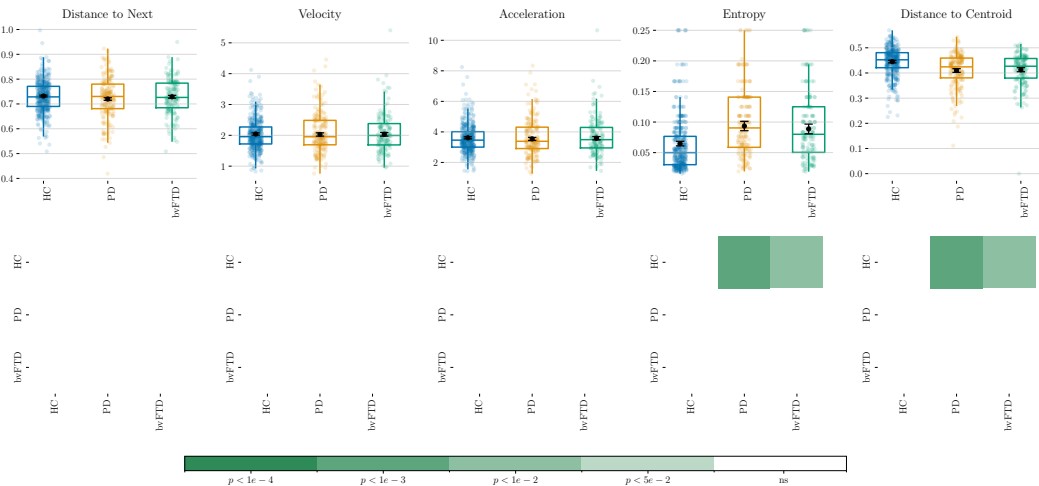

Figure 8: Neurodegenerative dataset, fastText (baseline; non-cumulative) model.

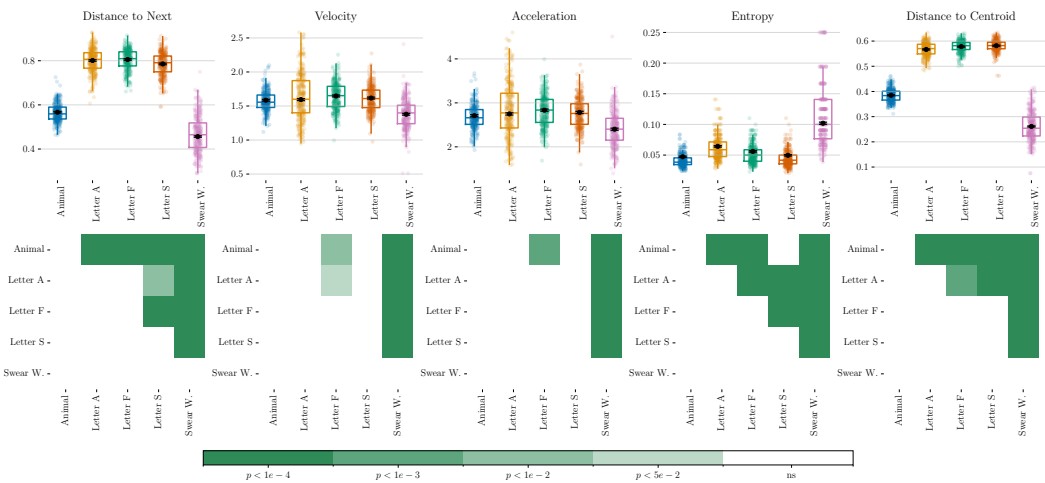

Figure 9: Swear Fluency dataset, fastText (baseline; non-cumulative) model.

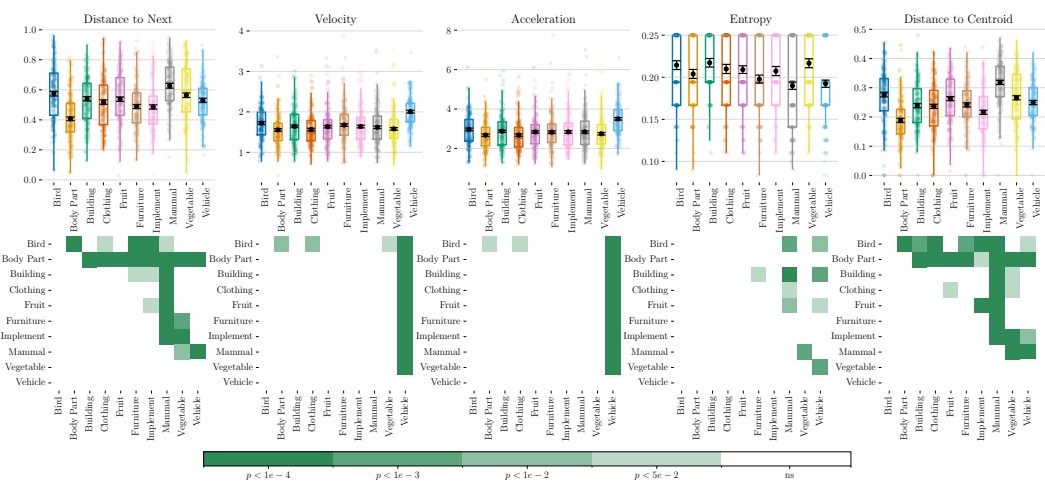

Figure 10: Italian dataset, fastText (baseline; non-cumulative) model.

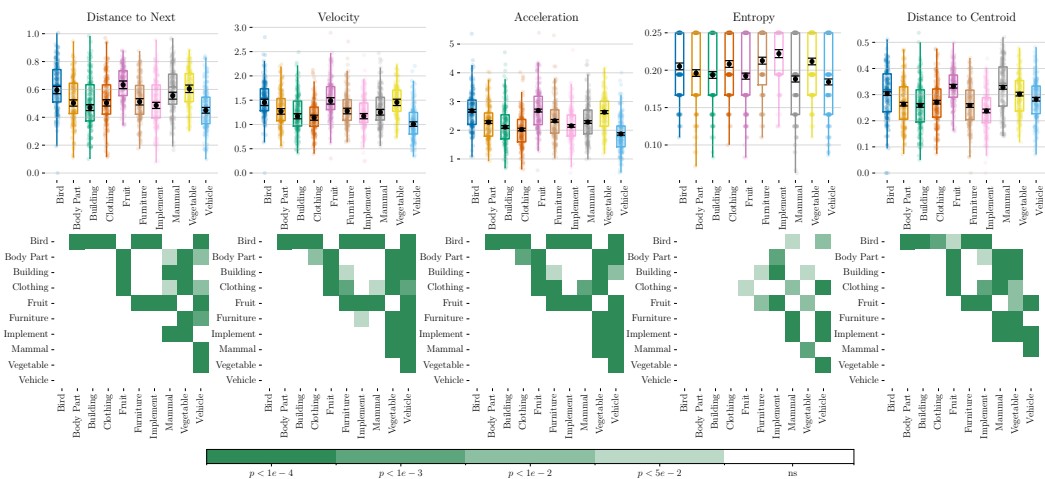

Figure 11: German dataset, fastText (baseline; non-cumulative) model.

## A.6 QWEN3-EMBEDDING-0.6B RESULTS

All experiments were reproduced for Qwen3-Embedding-0.6B, which is a small high-performance open-source embedding model (Zhang et al., 2025).

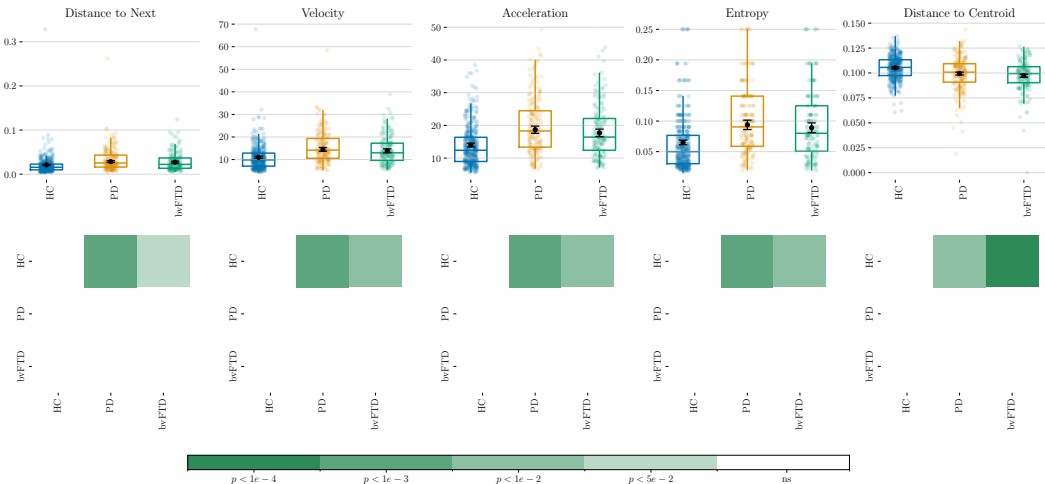

Figure 12: Neurodegenerative dataset, Qwen3-Embedding-0.6B model.

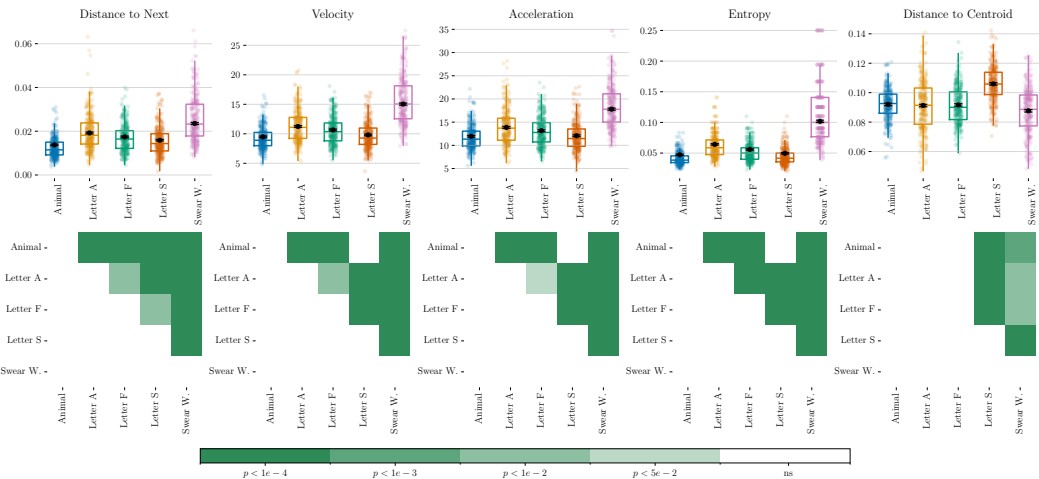

Figure 13: Swear Fluency dataset, Qwen3-Embedding-0.6B model.

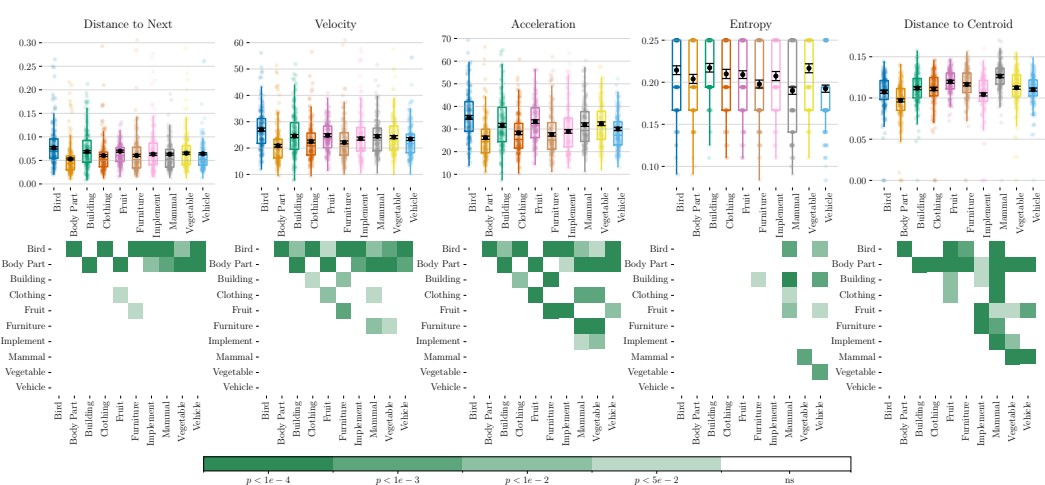

Figure 14: Italian dataset, Qwen3-Embedding-0.6B model.

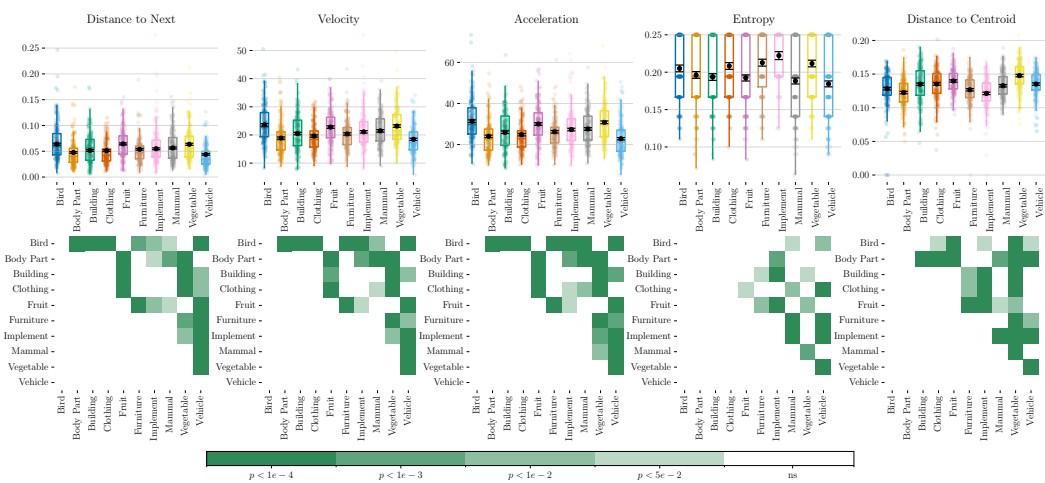

Figure 15: German dataset, Qwen3-Embedding-0.6B model.

## A.7 GOOGLE'S TEXT-EMBEDDING-004 RESULTS

All experiments were reproduced for Google's text-embedding-004.

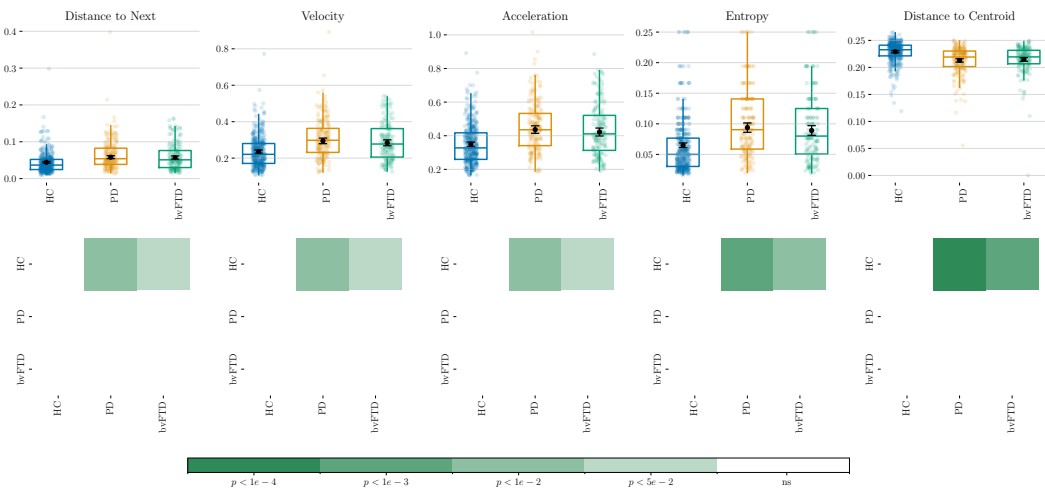

Figure 16: Neurodegenerative dataset, Google's text-embedding-004 model.

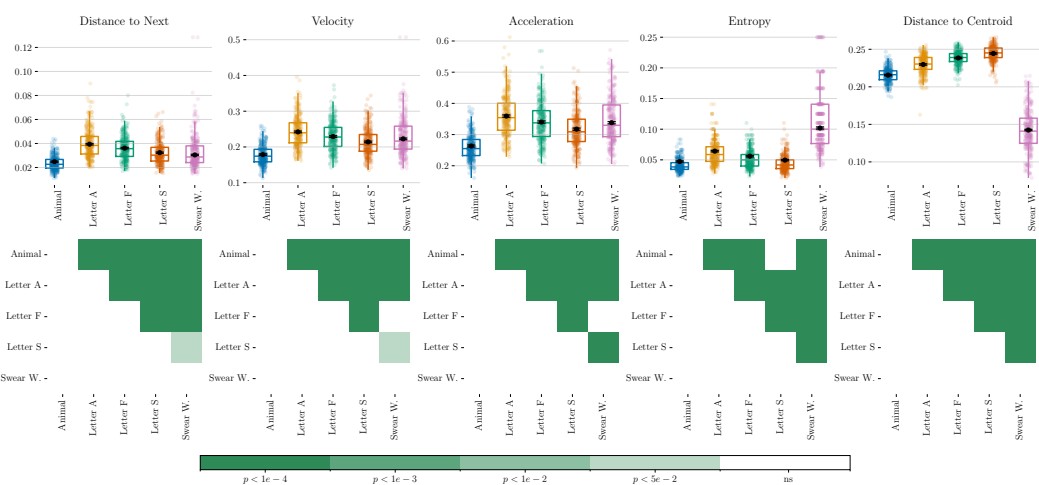

Figure 17: Swear Fluency dataset, Google's text-embedding-004 model.

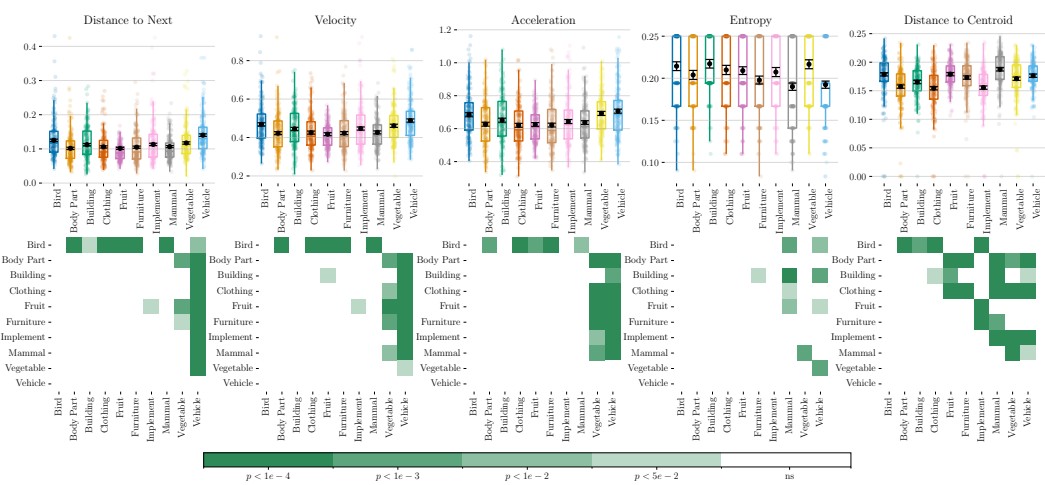

Figure 18: Italian dataset, Google's text-embedding-004 model.

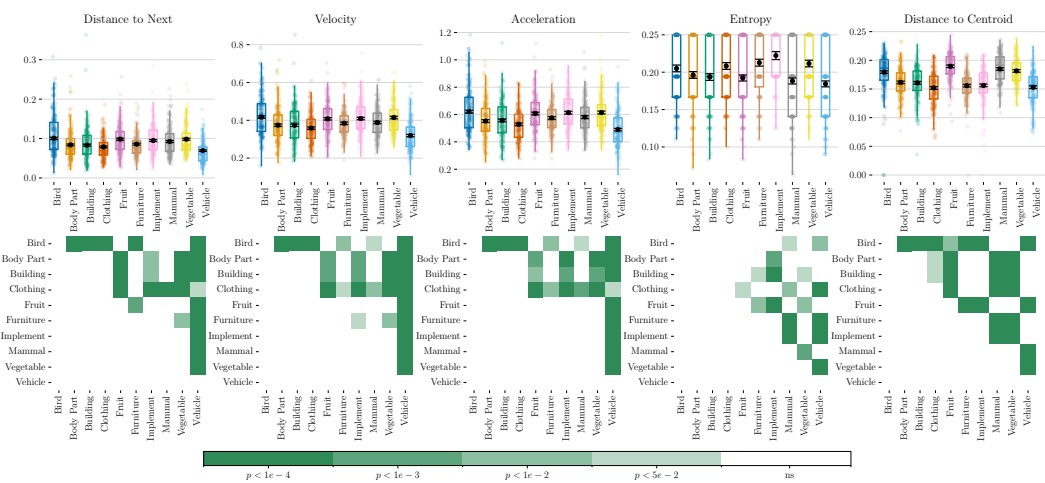

Figure 19: German dataset, Google's text-embedding-004 model.

