# OpenReview forum: "Characterizing Human Semantic Navigation in Concept Production as Trajectories in Embedding Space"
_ICLR.cc/2026/Conference — ICLR 2026 Poster_

### Official Review · Reviewer_GZZr · 2025-10-30

**Soundness:** 2
**Presentation:** 3
**Contribution:** 3
**Rating:** 6
**Confidence:** 3

**Summary:**

The paper models human concept production as trajectories in embedding space, extracting many geometric/dynamical markers from participant-specific paths built with cumulative text embeddings. The framework differentiates groups and concept types and appears robust across encoders for local trajectory metrics, while centroid-based dispersion shows model-dependent geometry. The authors discuss clinical and cross-lingual applications and note limitations.

**Strengths:**

Originality. Recasts verbal fluency/property listing as geometry + dynamics in learned representation spaces, bridging cognitive foraging accounts with modern NLP embeddings. The cumulative-embedding design captures history dependence rather than treating items independently.

Quality. Clear metric definitions (including a binarized entropy proxy) and mixed-effects modeling (GLMM via glmmTMB) are appropriate for repeated-measures data; cross-encoder replication is a strong robustness check.

Clarity. The paper’s pipeline is easy to follow; per-dataset result summaries and heatmaps for cross-model correlations aid interpretation.

Significance. Demonstrates clinically and linguistically informative signals with minimal manual annotation, and highlights encoder-agnostic local dynamics vs model-dependent global geometry, aligning with known cross-lingual structure and anisotropy issues in embeddings.

**Weaknesses:**

1. The analyses only adopt Euclidean differencing despite acknowledged anisotropy in contextual embeddings; consider non-Euclidean or locally whitened metrics (e.g., hyperbolic/Riemannian, subspace-projected velocities) to test robustness will be helpful.
2. Velocity/acceleration use implicit $\Delta t=1$ because timestamps are missing. Include more results to show where real inter-response times modulate the dynamics will be helpful.
3. Provide confidence intervals, seed/split variability, and multiple-comparison controls for pairwise tests in the figures; this will improve clinical interpretability.
4. Writing/format nits. A few typos (“Neurodegerative”), minor grammatical errors like “This approach hold…”; ensure all acronyms expand on first use.

**Questions:**

1. Do your main findings persist under whitened cosine, Riemannian distances, or other nonlinear metrics? A small ablation would help separate method from metric.
2. If inter-response times become available, do velocity/acceleration still distinguish groups once scaled by real time?
3. How do your metrics correlate with clustering/switching scores on the same sessions, and do they add incremental predictive value?
4. Given stable local dynamics across encoders, can you leverage multilingual alignment (e.g., shared subspaces) to standardize centroid-based dispersion across languages/models?

**Details Of Ethics Concerns:**

No ethics concern.

---

> ### Author Response · Authors · 2025-11-23
> **Response to Reviewer GZZr (1)**
>
> > W1: The analyses only adopt Euclidean differencing despite acknowledged anisotropy in contextual embeddings; consider non-Euclidean or locally whitened metrics (e.g., hyperbolic/Riemannian, subspace-projected velocities) to test robustness will be helpful.
>
> A-W1: We kindly ask the reviewer to read the answer to this in the next response (A-Q1).
>
> > W2: Velocity/acceleration use implicit $\Delta t=1$ because timestamps are missing. Include more results to show where real inter-response times modulate the dynamics will be helpful.
>
> A-W2: We kindly ask the reviewer to read the answer to this in the next response (A-Q2).
>
> > W3: Provide confidence intervals, seed/split variability, and multiple-comparison controls for pairwise tests in the figures; this will improve clinical interpretability.
>
> A-W3: We appreciate the reviewer’s statistical rigor to enhance clinical interpretability, and we believe these elements are addressed in our analysis and visualizations. Our approach accounts for uncertainty and multiple testing. Our Generalized Linear Mixed Models (GLMMs) provided model-estimated marginal means and 95% Confidence Intervals, which are visualized as black point-ranges overlaid on the raw data distributions in all figures (please see Section 2.2.2). Furthermore, all pairwise comparisons were strictly corrected using Tukey’s HSD to control the family-wise error rate. Regarding the display of these tests, we opted for significance matrices positioned directly below the distributions rather than using overhead bars; given that some datasets involve up to 10 categories (resulting in 45 possible pairwise contrasts), adding individual bracket lines would have created excessive visual clutter, whereas the matrices provide a comprehensive and readable map of all corrected significant differences. Finally, regarding seed/split variability, in our statistical approach, these are accounted for through Random Effects within the mixed models, allowing for valid population inferences without reducing statistical power through data splitting.
>
> > W4: Writing/format nits. A few typos (“Neurodegerative”), minor grammatical errors like “This approach hold…”; ensure all acronyms expand on first use.
>
> A-W4: Thank you for your careful reading. We did a careful revision, looking for typos and grammatical errors, and fixed them.
>
> ---
>
> > Q1: Do your main findings persist under whitened cosine, Riemannian distances, or other nonlinear metrics? A small ablation would help separate method from metric.
>
> A-Q1 (& A-W1): We agree with you and the reviewers who raised this problem. We decided to address the embedding anisotropy by applying a ZCA-whitening transformation to our embeddings and recalculated the metrics to assess the robustness of group discrimination under more approximated isotropic conditions (Bell, 1997; Su et al., 2021; Zhuo et al., 2023). These results showed only small changes in group differences. We added this in the methods and limitations sections; the results are briefly discussed in the Appendix section A.4.
>
> > Q2: If inter-response times become available, do velocity/acceleration still distinguish groups once scaled by real time?
>
> A-Q2 (& A-W2): Yes, we believe it would add valuable information and make the metrics even more similar to the physics dynamics. However, most datasets don't include this information, mostly due to complications in the experimental design to include trustworthy time stamps. But we included in the text a remark regarding this and how future experiments can incorporate this information in the Limitation and Future Work section.

---

> ### Author Response · Authors · 2025-11-23
> **Response to Reviewer GZZr (2)**
>
> > Q3: How do your metrics correlate with clustering/switching scores on the same sessions, and do they add incremental predictive value?
>
> A-Q3: We appreciate this insightful question. While we did not perform a direct statistical correlation analysis in this study, we frame our metrics as a continuous, high-resolution extension of the discrete clustering and switching paradigm (Troyer et al., 1997). Conceptually, high distance to next, velocity, and acceleration serve as geometric proxies for the "switching" mechanism, while low values correspond to "clustering" or exploitation. Entropy complements this by indexing the global irregularity of the search strategy, distinguishing between predictable, routine traversals and disordered navigation, while distance to centroid captures the overall spatial dispersion of the search, a topological dimension that discrete counts do not explicitly measure. The incremental value of our approach lies in this granularity: whereas traditional scores provide a binary count of transitions, our kinematic and geometric metrics quantify the magnitude, volatility, and spatial extent of these processes step-by-step. This allows us to distinguish between a "stable" switch (strategic exploration) and an "erratic" switch (executive instability), or to detect a constricted semantic space, nuances that binary counts may miss. We have revised the Introduction to explicitly position our framework as capturing this "step-by-step granularity" that complements traditional measures, and the Discussion to interpret these dynamics as specific signatures of executive function, thereby adding mechanistic explanatory value beyond simple predictive classification.
>
> > Q4: Given stable local dynamics across encoders, can you leverage multilingual alignment (e.g., shared subspaces) to standardize centroid-based dispersion across languages/models?
>
> A-Q4: We value your question, but despite the local stable dynamics across encoders within datasets, results show some differences between Italian and German datasets, as shown by figures 4 and 5, suggesting a different dynamic across languages. Thus, we could in principle try to standardize the centroid-based dispersion, but in this present work, we are more interested in the non-standardization and look into the linguistic differences. Thus, due to your comment, we added a discussion of this multilingual difference in paragraph 4 in the Discussion section.

---

> > ### Author Response · Authors · 2025-11-23
> > **References**
> >
> > Bell, A. J., & Sejnowski, T. J. (1997). The “independent components” of natural scenes are edge filters. Vision Research, 37(23), 3327–3338. https://doi.org/10.1016/S0042-6989(97)00121-1
> >
> > Troyer, A. K., Moscovitch, M., & Winocur, G. (1997). Clustering and switching as two components of verbal fluency: Evidence from younger and older healthy adults. Neuropsychology, 11(1), 138–146. https://doi.org/10.1037/0894-4105.11.1.138
> >
> > Su, J., Cao, J., Liu, W., & Ou, Y. (2021). Whitening sentence representations for better semantics and faster retrieval. arXiv. https://doi.org/10.48550/arXiv.2103.15316
> >
> > Zhuo, W., Sun, Y., Wang, X., Zhu, L., & Yang, Y. (2023). WhitenedCSE: Whitening-based contrastive learning of sentence embeddings. In A. Rogers, J. Boyd-Graber, & N. Okazaki (Eds.), Proceedings of the 61st Annual Meeting of the Association for Computational Linguistics (Volume 1: Long Papers) (pp. 12135–12148). Association for Computational Linguistics. https://doi.org/10.18653/v1/2023.acl-long.677

---

### Official Review · Reviewer_Xfqo · 2025-10-30

**Soundness:** 3
**Presentation:** 3
**Contribution:** 3
**Rating:** 6
**Confidence:** 4

**Summary:**

*Concept production* refers to the task of a person listing as many words as they can within a given category in a short time period. This paper proposes a method to quantify the semantic dynamics of concept production by measuring dynamic properties of Language Model embeddings of the produced words.

They authors propose 5 summary statistics of the embedding trajectories: distance-to-next, entropy, velocity, acceleration and distance-to-centroid. They employ four datasets of to evaluate:
 - a neurodegenerative dataset containing participants with Parkinson's disease, frontotemporal dementia and healthy controls,
- a swear fluency dataset, with three categories control categories and I swear word category,
 - Italian and German datasets, where particants were asked to produce words from a variety to categories in either Italian or German

Qualitative results in the form of plots, and quantitative statistical test are performed to assess the variability between categories of the embedding dynamics within datasets.

The authors perform their experiment using three different embedding models and find similar results.

**Strengths:**

The paper is clearly written and the methodology is novel, providing a means to quantify properties of the semantic dynamics of words produced during these experiment using Language model embeddings as a proxy.

The paper considers a variety of tasks, datasets and models and the results are clearly explained.

**Weaknesses:**

While this work clearly shows how the metrics they propose vary across groups in their datasets,  it's not clear to me what this actually tells us about human semantic navigation.

I think there is some value in the proposed metrics, but the experiments only provide a weak indication that they are useful for the classification tasks described.

**Questions:**

- Does the findings here align with other research on human semantic cognition?
- What other methods exist to measure the semantic dynamics of words produced in these experiments? For example. what metrics are used in the papers originally proposing the datasets? You mention that your finding corroborate theirs, but how?
- What are the advantages of this method over other methods?
- In the definition of entropy, the time series $\{x_t\}_{t=1}^n$ is a vector time series, so what is the median of a set of vectors?
- Similarly, the velocity and acceleration are vectors. Do you report their magnitude as the metric, or something else?

---

> ### Author Response · Authors · 2025-11-23
> **Response to Reviewer Xfqo (1)**
>
> Thank you very much for your review and your kind comments regarding our work. They have been very insightful for the improvement of our manuscript.
>
> > W: While this work clearly shows how the metrics they propose vary across groups in their datasets, it's not clear to me what this actually tells us about human semantic navigation.
>
> I think there is some value in the proposed metrics, but the experiments only provide a weak indication that they are useful for the classification tasks described.
>
> A-W: We appreciate the reviewer’s constructive feedback regarding the interpretability and utility of our findings in the context of human semantic navigation. We agree that the link between our geometric metrics and established cognitive theories needed strengthening to fully demonstrate their value beyond simple classification features. To address this, we have strengthened the cognitive explanations throughout the manuscript. In the Introduction (paragraphs 2-3), we now explicitly address how our metrics characterize human semantic navigation through the lens of executive functions, explaining how working memory and inhibition necessitate the cumulative modeling approach we propose. In the Methods section, we added clearer cognitive interpretations for each metric (e.g., linking acceleration to switching instability) to aid in understanding how they are useful for characterizing specific aspects of navigation. Finally, we substantially improved the Discussion to interpret our results in terms of specific cognitive differences among groups and concept types, such as identifying signatures of executive dysfunction and diminished sensorimotor traces in patients. These revisions demonstrate that our metrics provide a mechanistic, computationally grounded view of semantic retrieval that offers insights far beyond classifying or differentiating populations.
>
> ---
>
> > Q-1: Does the findings here align with other research on human semantic cognition?
>
> A-Q1: Thank you for your question, which allows us to clarify the importance of our results and the relevance of our method. Yes, our findings align closely with and extend established research in human semantic cognition across three key domains. First, in the context of neurodegeneration, the high velocity, acceleration, and entropy observed in Parkinson’s and bvFTD patients corroborate the well-documented executive dysfunction and deficits in strategic retrieval (clustering/switching) characteristic of these populations (Birba et al., 2017; Cousins & Grossman, 2017; Toro-Hernández et al., 2024). Furthermore, the reduced distance to centroid in these groups aligns with theories positing a "diminished semantic space" (Zhang, et al 2022) resulting from the degradation of sensorimotor simulations required to ground complex concepts (Fernandino et al., 2013; Speed et al., 2017). Second, regarding taboo lexico-semantics, our finding that swear word fluency involves high kinematic volatility within a spatially centralized neighborhood supports models that distinguish the organization of taboo words from standard taxonomic categories (Jay & Jay, 2015). Third, our cross-linguistic results (Italian vs. German) align with theories of linguistic relativity and grounded cognition, which suggest that structural differences in language (e.g., how specificity and concreteness are encoded) shape the trajectory of semantic access (Vigliocco et al., 2009; Bolognesi et al., 2020; Barsalou, 2023). Collectively, our framework provides a novel geometric quantification of these known cognitive phenomena. We discussed these results in the Discussion section, in paragraphs 2,3, and 4.

---

> > ### Author Response · Authors · 2025-11-23
> > **Response to Reviewer Xfqo (2)**
> >
> > > Q2//Q3: What other methods exist to measure the semantic dynamics of words produced in these experiments? For example. what metrics are used in the papers originally proposing the datasets? You mention that your finding corroborate theirs, but how?//What are the advantages of this method over other methods?
> >
> > A-Q2//Q3: Traditional methods for analyzing these tasks rely on discrete metrics such as word counts, clustering/switching scores, or graph measures. Our findings not only corroborate these studies but also provide complementary approaches by modeling the inherently cumulative process of semantic navigation as a continuous geometric trajectory. This yields an automated, high-resolution validation of established behavioral phenomena. For example, our results confirm previous natural language processing analyses of semantic variability in Parkinson’s disease (Toro-Hernández, et al., 2024), providing step-by-step granularity to the interpretation of navigation. Similarly, our distinct geometric profiles for Italian and German speakers validate prior observations of qualitative differences in semantic organization, even when trajectory lengths are similar (Kremer, and Baroni, 2011).
> >
> > To validate our framework's added value through benchmarking, we introduced a non-cumulative fastText (Mikolov et al., 2018) baseline, which directly replicates the traditional pairwise approach (Linz et al., 2017) (see Table 2 and Appendix A.6). This quantitative comparison reveals a clear length-dependent advantage: while the non-cumulative fastText baseline excelled with short word lists (Italian/German), our proposed cumulative transformer approach significantly outperformed all non-cumulative baselines when analyzing the longer trajectories found in the Neurodegenerative dataset. Crucially, the introduced metrics are designed to be complementary to prior literature, capturing distinct dimensions of semantic navigation: semantic change (distance to next), magnitude of local dynamics (velocity and acceleration), predictability of searching process (entropy), and dispersion or “closedness” to central meaning (distance to centroid).
> >
> > > Q4: In the definition of entropy, the time series  is a vector time series, so what is the median of a set of vectors?
> >
> > A-Q4: Sorry for the confusion, the entropy evaluation in the time series is not on a vector time series. This metric is derived by classifying distances as "high" or "low" relative to the distance-to-next trajectory's median, then calculating the Shannon entropy of this binary sequence normalized by the number of valid steps. We have revised and modified the Methods section accordingly.
> >
> > > Q5: Similarly, the velocity and acceleration are vectors. Do you report their magnitude as the metric, or something else?
> >
> > A-Q5: Thank you for the careful read. For reporting scalar metrics, we use the magnitudes of the vectors (i.e., Euclidean norms). We updated the methods section to make it clearer.

---

> > > ### Author Response · Authors · 2025-11-23
> > > **References**
> > >
> > > Barsalou, L. W. (2023). Implications of grounded cognition for conceptual processing across cultures. Topics in Cognitive Science, 15(4), 648-656.
> > >
> > > Birba, A., García-Cordero, I., Kozono, G., Legaz, A., Ibáñez, A., Sedeño, L., & García, A. M. (2017). Losing ground: Frontostriatal atrophy disrupts language embodiment in Parkinson's and Huntington's disease. Neuroscience & Biobehavioral Reviews, 80, 673–687. https://doi.org/10.1016/j.neubiorev.2017.07.011
> > >
> > > Bolognesi, M., Burgers, C., & Caselli, T. (2020). On abstraction: decoupling conceptual concreteness and categorical specificity. Cognitive Processing, 21(3), 365-381.
> > >
> > > Cousins, K. A. Q., & Grossman, M. (2017). Evidence of semantic processing impairments in behavioural variant frontotemporal dementia and Parkinson's disease. Current opinion in neurology, 30(6), 617–622. https://doi.org/10.1097/WCO.0000000000000498
> > >
> > > Fernandino, L., Conant, L. L., Binder, J. R., Blindauer, K., Hiner, B., Spangler, K., & Desai, R. H. (2013). Parkinson’s disease disrupts both automatic and controlled processing of action verbs. Brain and Language, 127(1), 65–74.
> > >
> > > Jay, K. L., & Jay, T. B. (2015). Taboo word fluency and knowledge of slurs and general pejoratives: Deconstructing the poverty-of-vocabulary myth. Language Sciences, 52, 251-259.
> > >
> > > Kremer, G., & Baroni, M. (2011). A set of semantic norms for German and Italian. Behavior Research Methods, 43(1), 97-109.
> > >
> > > Linz, N., Tröger, J., Alexandersson, J., & König, A. (2017). Using neural word embeddings in the analysis of the clinical semantic verbal fluency task. In Proceedings of the 12th International Conference on Computational Semantics (IWCS)—Short papers.
> > >
> > > Mikolov, T., Grave, E., Bojanowski, P., Puhrsch, C., & Joulin, A. (2018). Advances in Pre-Training Distributed Word Representations. In Proceedings of the International Conference on Language Resources and Evaluation (LREC 2018).
> > >
> > > Speed, L. J., van Dam, W. O., Hirath, P., Vigliocco, G., & Desai, R. H. (2017). Impaired comprehension of speed verbs in Parkinson’s disease. Journal of the International Neuropsychological Society, 23(5), 412–420.
> > >
> > > Toro-Hernández, F. D., Migeot, J., Marchant, N., Olivares, D., Ferrante, F., González-Gómez, R., González Campo, C., Fittipaldi, S., Rojas-Costa, G. M., Moguilner, S., Slachevsky, A., Chaná Cuevas, P., Ibáñez, A., Chaigneau, S., & García, A. M. (2024). Author Correction: Neurocognitive correlates of semantic memory navigation in Parkinson's disease. NPJ Parkinson's disease, 10(1), 30. https://doi.org/10.1038/s41531-024-00644-y
> > >
> > > Vigliocco, G., Meteyard, L., Andrews, M., & Kousta, S. (2009). Toward a theory of semantic representation. Language and Cognition, 1(2), 219-247.
> > >
> > > Zhang, G., Ma, J., Chan, P., & Ye, Z. (2022). Graph Theoretical Analysis of Semantic Fluency in Patients with Parkinson's Disease. Behavioural neurology, 2022, 6935263. https://doi.org/10.1155/2022/6935263

---

> > > > ### Comment · Reviewer_Xfqo · 2025-11-24
> > > >
> > > > Thank you for your considered response. I continue to recommend marginal acceptance of this paper.

---

### Official Review · Reviewer_6H8S · 2025-10-30

**Soundness:** 3
**Presentation:** 2
**Contribution:** 2
**Rating:** 2
**Confidence:** 3

**Summary:**

This paper proposes a framework to quantify human semantic navigation during concept production using trajectory-based metrics in transformer embedding spaces. It represents sequential concept generation (e.g., verbal fluency or property listing tasks) as a path through semantic space and computes geometric and dynamical metrics—distance to next, entropy, velocity, acceleration, and distance to centroid—to characterize this navigation. The method is applied to four datasets (neurodegenerative, swear-word fluency, and Italian/German property listing), showing that these metrics distinguish clinical groups, semantic categories, and languages. Results are robust across embedding models (OpenAI, Google, Qwen). The authors argue this approach bridges cognitive modeling and learned representations, offering potential applications in clinical and cross-linguistic research.

Overall, the approach is interesting. However, its mathematical grounding is limited, and comparisons to prior methods are absent. It is unclear whether this paper is primarily a methodological contribution, or an empirical contribution. While it seems to be a mixture of both, neither is particularly compelling. This work may be more suitable as an expanded manuscript for a cognitive science journal venue, where the results can be more effectively situated within the broader relevant literature, and targeted towards a more appropriate community.

**Strengths:**

1. Potentially novel conceptual framing: the paper presents a method for modeling semantic search as trajectories in embedding space, bridging computational linguistics and cognitive science.
2. Methodological simplicity and reproducibility: the framework requires minimal manual annotation and is implemented with publicly available datasets and embeddings, making it scalable and easy to replicate.
3. Comprehensive empirical validation: the authors test across multiple datasets (languages, clinical populations, and semantic domains), providing strong evidence of the method’s generality.
4. Cross-model robustness analysis: the inclusion of multiple embedding models (OpenAI, Google, Qwen) and the correlation analysis (Figure 6) convincingly demonstrate stability of local trajectory measures across model architectures.
5. Interdisciplinary contribution: the work effectively connects semantic cognition, NLP, and neuropsychology, potentially valuable for both cognitive modeling and clinical diagnostics.

**Weaknesses:**

1. Limited theoretical grounding for metrics: while the chosen metrics (velocity, acceleration, entropy) are intuitive, their psychological interpretation is underspecified. The link between these geometric quantities and cognitive mechanisms of search (e.g., clustering/switching, semantic control) could be better formalized.
2. Simplistic dynamics assumption: the framework assumes Euclidean dynamics, even though embeddings are anisotropic and often non-Euclidean. The authors acknowledge this multiple times, but do not explore or justify why Euclidean treatment suffices.
3. Use of non-causal embeddings. It is a reasonable first step to use cumulative text embeddings, rather than independent word embeddings. However, by using non-causal encoder models for embeddings, for a sequence "A B C", the representation of token B has access to token "C", whereas in the earlier part of the sequence it does not. Rather than acquiring embeddings, the authors should use a causally-masked model, such that the token representation for "B" is the same across both "A B C" and "A B". This would likely make the trajectories smoother and more amenable to the kinematic metrics employed.
4. Figures are dense: Figures 2–5 (and corresponding appendices) show many small boxplots and correlation matrices. It may be helpful to use comparison lines above the scatter plots rather than the correlation matrices, if the goal is merely to show which effects are significant. This is a more standard approach and would take less space, improving readability.
5. No comparison to traditional linguistic baselines: the framework is presented as superior to “labor-intensive linguistic pre-processing,” yet there’s no quantitative comparison to classical measures (e.g., clustering, switching, word frequency, semantic similarity). A comparison with these approaches would strengthen the claim of added value.
6. Missing temporal information: since no timestamp data are used, “velocity” and “acceleration” are only metaphorical. The interpretation of these measures as cognitive dynamics rather than geometric derivatives is thus limited.
7. The work only characterizes semantic trajectories, it does not model them through some sampling from the latent space of a transformer. This makes the theoretical contribution weaker.

**Questions:**

1. How does this work improve upon prior baselines?
2. Why are all of the metrics needed? What are the main distinctions between them?
3. Do you agree with weakness 3? Can you perform another analysis with a causally masked transformer?

---

> ### Author Response · Authors · 2025-11-23
> **Response to Reviewer 6H8S (1)**
>
> Dear reviewer,
>
> Thank you for your careful revision of our manuscript. We will use your comments and suggestions to better clarify the relevance of our approach and highlight its computational impacts, regarding applicability not only in cognitive neuroscience but also in computer science.
>
> We deeply believe that cognitive and learning representations have an intimate relationship, since both disciplines can benefit from this interaction as they grow together, bridging gaps between human and artificial cognition. For this reason, we propose a characterization of human semantic navigation through the lens of learned language representations over different models and datasets.
>
> > W1: Limited theoretical grounding for metrics: while the chosen metrics (velocity, acceleration, entropy) are intuitive, their psychological interpretation is underspecified. The link between these geometric quantities and cognitive mechanisms of search (e.g., clustering/switching, semantic control) could be better formalized.
>
> A-W1: Thank you for your comments on this. We agree on the need to further establish the link between metrics and cognitive mechanisms of search. To address it, we better formalized the relationship between cognitive mechanisms on semantic memory search and their use with the embedding models methodology. We further discuss this in the Introduction section, especially in paragraphs 2 and 3. Also, we added clarification for each metric explanation at the end of each corresponding paragraph in the Methods section (see “2.2 Characterizing navigation”), and added a specific paragraph summarizing all explanations to close it. We now have explicitly linked our trajectory dynamics in the classical terms of exploitation and exploration, used in the literature of clustering and switching, to understand how our metrics extend upon them.
>
> > W2: Simplistic dynamics assumption: the framework assumes Euclidean dynamics, even though embeddings are anisotropic and often non-Euclidean. The authors acknowledge this multiple times, but do not explore or justify why Euclidean treatment suffices.
>
> A-W2: We agree with you and the reviewers who raised this problem. We decided to address the embedding anisotropy by applying a ZCA-whitening transformation to our embeddings and recalculated the metrics to assess the robustness of group discrimination under more approximated isotropic conditions (Bell, 1997; Su et al., 2021; Zhuo et al., 2023). These results showed only small changes in group differences. We added this in the methods and limitations sections; the results are briefly discussed in the Appendix section A.4.
>
> > W3: Use of non-causal embeddings. It is a reasonable first step to use cumulative text embeddings, rather than independent word embeddings. However, by using non-causal encoder models for embeddings, for a sequence "A B C", the representation of token B has access to token "C", whereas in the earlier part of the sequence it does not. Rather than acquiring embeddings, the authors should use a causally-masked model, such that the token representation for "B" is the same across both "A B C" and "A B". This would likely make the trajectories smoother and more amenable to the kinematic metrics employed.
>
> A-W3: We kindly ask the reviewer to read the answer to this in the next response (A-Q3)
>
> > W4: Figures are dense: Figures 2–5 (and corresponding appendices) show many small boxplots and correlation matrices. It may be helpful to use comparison lines above the scatter plots rather than the correlation matrices, if the goal is merely to show which effects are significant. This is a more standard approach and would take less space, improving readability.
>
> A-W4: We agreed that the figures needed clarity for a more careful reading. Our first approach used comparison lines above the boxplots, but as some datasets have a lot of comparisons, the results were unreadable. For that reason, we decided to keep the main structure of the plot since it may provide an insightful highlight to the comparisons, particularly for the cognitive neuroscience community. Then, we adapt them to ease readability by cleaning them (improved spacing between plots, removed the asterisks for cleanness, and enhanced font).
>
> > W5: No comparison to traditional linguistic baselines: the framework is presented as superior to “labor-intensive linguistic pre-processing,” yet there’s no quantitative comparison to classical measures (e.g., clustering, switching, word frequency, semantic similarity). A comparison with these approaches would strengthen the claim of added value.
>
> A-W5: We kindly ask the reviewer to read the answer to this in the next response (A-Q1)

---

> > ### Author Response · Authors · 2025-11-23
> > **Response to Reviewer 6H8S (2)**
> >
> > > W6: Missing temporal information: since no timestamp data are used, “velocity” and “acceleration” are only metaphorical. The interpretation of these measures as cognitive dynamics rather than geometric derivatives is thus limited.
> >
> > A-W6: We further agree that timestamps are pivotal to better characterize semantic memory dynamics from a physics perspective. However, this data is not easily found in open datasets, which makes it harder to perform cross-data analyses that can be compared. But we included in the text a remark regarding this and how future experiments can incorporate this information. We address this in the Limitations and Future Work section.
> >
> > > W7: The work only characterizes semantic trajectories, it does not model them through some sampling from the latent space of a transformer. This makes the theoretical contribution weaker.
> >
> > A-W7: The reviewer raises a valid point. Our work so far has been descriptive and analytical; we quantify what the human trajectories look like, but we did not include an explicit generative or mechanistic model of how those trajectories might be produced. We agree that a generative model (e.g., sampling from a transformer’s latent space) would enhance the theoretical contribution. However, we maintain that precise empirical characterization is a necessary prerequisite to mechanistic modeling. Our current study focuses on establishing these quantitative baselines. We have revised the Discussion to explicitly address this limitation, outlining how our proposed metrics lay the groundwork for future work comparing human trajectories against artificial agents and random walk models. Your comment inspired us to pursue another project in the exploration of latent space for concept production using different sampling methods.
> >
> > ---
> >
> > > Q1: How does this work improve upon prior baselines?
> >
> > A-Q1 (& A-W5): We thank the reviewer for highlighting the importance of benchmarking our framework against established methods to strengthen the claim of added value. To address this quantitatively, we introduced a non-cumulative fastText (Mikolov et al., 2018) baseline condition to directly replicate the traditional pairwise approach used in prior work like Linz et al. (2017), detailed in Table 2 and Appendix A.6. Our analysis revealed a distinct dissociation based on trajectory length: while traditional non-cumulative metrics yielded stronger effect sizes for short word lists (Italian/German), our proposed cumulative transformer approach significantly outperformed non-cumulative baselines, including fastText, in the Neurodegenerative dataset. This indicates that modeling the cumulative search history is critical for capturing the broader semantic structure and disorganized patterns found in clinical groups producing longer trajectories. Furthermore, regarding the theoretical added value, we agree that the link between our geometric metrics and established cognitive theories needed strengthening. We have rewritten the Introduction paragraphs 2 and 3 to explicitly frame our approach as an extension of the clustering/switching paradigm (Troyer et al., 1997) and foraging models (Hills et al., 2012). Specifically, we now posit that our kinematic metrics (velocity, acceleration) capture the "step-by-step granularity" of the search process, offering a continuous view that complements the binary nature of clustering and switching. This complementarity helps us interpret different characterizations of semantic memory, easing group comparison and the understanding of the cognitive mechanisms behind them, as further discussed in the revised Discussion section, paragraphs 2, 3, and 4.
> >
> > > Q2: Why are all of the metrics needed? What are the main distinctions between them?
> >
> > A-Q2: Thank you for your question. We propose a set of physics-based metrics, specifically distance-to-next, velocity, and acceleration, that together describe the navigation dynamics. These are complemented by entropy for informational assessment and distance-to-centroid for global geometry. We address this now specifically in the text, in which we further differentiate each metric at the end of each description (see “2.2 Characterizing navigation” in the Methods section). Also, we added a new paragraph at the end of the Methods section. We have clarified that the metrics are designed to be complementary, covering different dimensions of semantic navigation, such as step-by-step variation of semantic content (distance to next), magnitude of local kinematics (velocity and acceleration), predictability of searching process (entropy), and dispersion or “closedness” to central meaning (distance to centroid).

---

> > > ### Author Response · Authors · 2025-11-23
> > > **Response to Reviewer 6H8S (3)**
> > >
> > > > Q3: Do you agree with weakness 3? Can you perform another analysis with a causally masked transformer?
> > >
> > > A-Q3: Thank you for the thoughtful suggestion. We agree that causally masked representations can, in principle, make prefix‑wise trajectories more plausible and cognitive accurate. For this we used three models trained differently. Despite these architectural differences (causal vs. bidirectional attention, and one unspecified), our trajectory-level and kinematic metrics were consistent across models, and we did not observe material changes attributable to causality. We added the model information into the methods section.
> > > Qwen3-Embedding-0.6B (causal) input uses causal attention and the final vector is the last-layer hidden state at an appended [EOS] token. Thus, when we embed a prefix 1:t, the representation for t cannot attend to tokens >t (Zhang et al., 2025).
> > >
> > > Google text-embedding-004 (encoder-style) Google’s public papers describe two relevant encoder-style families: Gecko, a dual-encoder/bi-encoder retriever distilled from an LLM (text-embedding-004), and Gemini Embedding, which uses a bidirectional transformer with mean pooling. Both are non-causal encoders (Lee et al., 2024).
> > >
> > > OpenAI text-embedding-3 Architecture details are not publicly disclosed in official docs/catalogs; we therefore treat them as unspecified.

---

> ### Author Response · Authors · 2025-11-23
> **References**
>
> Bell, A. J., & Sejnowski, T. J. (1997). The “independent components” of natural scenes are edge filters. Vision Research, 37(23), 3327–3338. https://doi.org/10.1016/S0042-6989(97)00121-1
>
> Hills, T. T., Jones, M. N., & Todd, P. M. (2012). Optimal foraging in semantic memory. Psychological Review, 119(2), 431–440. https://doi.org/10.1037/a0027373
>
> Lee, J., Dai, Z., Ren, X., Chen, B., Cer, D., Cole, J. R., Hui, K., Boratko, M., Kapadia, R., Ding, W., Luan, Y., Duddu, S. M. K., Abrego, G. H., Shi, W., Gupta, N., Kusupati, A., Jain, P., Jonnalagadda, S. R., Chang, M.-W., & Naim, I. (2024). Gecko: Versatile text embeddings distilled from large language models. arXiv. https://doi.org/10.48550/arXiv.2403.20327
>
> Linz, N., Tröger, J., Alexandersson, J., & König, A. (2017). Using Neural Word Embeddings in the Analysis of the Clinical Semantic Verbal Fluency Task. In Proceedings of the 12th International Conference on Computational Semantics (IWCS) — Short Papers (pp. 1–7). The Association for Computer Linguistics. https://doi.org/10.18653/v1/W17-6926
>
> Mikolov, T., Grave, E., Bojanowski, P., Puhrsch, C., & Joulin, A. (2018). Advances in Pre-Training Distributed Word Representations. In Proceedings of the International Conference on Language Resources and Evaluation (LREC 2018).
>
> Su, J., Cao, J., Liu, W., & Ou, Y. (2021). Whitening sentence representations for better semantics and faster retrieval. arXiv. https://doi.org/10.48550/arXiv.2103.15316
>
> Troyer, A. K., Moscovitch, M., & Winocur, G. (1997). Clustering and switching as two components of verbal fluency: Evidence from younger and older healthy adults. Neuropsychology, 11(1), 138–146. https://doi.org/10.1037/0894-4105.11.1.138
>
> Zhuo, W., Sun, Y., Wang, X., Zhu, L., & Yang, Y. (2023). WhitenedCSE: Whitening-based contrastive learning of sentence embeddings. In A. Rogers, J. Boyd-Graber, & N. Okazaki (Eds.), Proceedings of the 61st Annual Meeting of the Association for Computational Linguistics (Volume 1: Long Papers) (pp. 12135–12148). Association for Computational Linguistics. https://doi.org/10.18653/v1/2023.acl-long.677
>
> Zhang, Y., Li, M., Long, D., Zhang, X., Lin, H., Yang, B., Xie, P., Yang, A., Liu, D., Lin, J., Huang, F., & Zhou, J. (2025). Qwen3 embedding: Advancing text embedding and reranking through foundation models (Version 3). arXiv. https://doi.org/10.48550/arXiv.2506.05176

---

### Official Review · Reviewer_xS4F · 2025-10-31

**Soundness:** 3
**Presentation:** 3
**Contribution:** 3
**Rating:** 6
**Confidence:** 4

**Summary:**

This paper proposes a framework for characterizing human semantic navigation by representing concept production tasks (semantic fluency and property listing) as trajectories through transformer-based embedding spaces. The authors extract geometric and dynamical metrics including distance-to-next, velocity, acceleration, entropy, and distance-to-centroid from cumulative word sequences. They evaluate their approach on four datasets spanning clinical populations (Parkinson's, frontotemporal dementia), different languages (Italian, German), and semantic categories, showing that these trajectory-based metrics can distinguish between clinical groups and concept types across different transformer models.

**Strengths:**

- Novel computational framework: The trajectory-based approach to semantic navigation is creative, moving beyond static embedding analyses to capture dynamic aspects of semantic search. The use of cumulative embeddings (where x_t encodes items 1:t) is particularly interesting as it captures sequential dependencies.
- Robust empirical validation: The evaluation across four diverse datasets (clinical, multilingual, different task types) demonstrates broad applicability. The consistency of findings across three different embedding models (OpenAI, Google, Qwen) strengthens the results.
- Clinical relevance: Successfully differentiating neurodegenerative groups from healthy controls using distance-to-next and other metrics provides potential clinical utility. The finding that patient groups show greater variability and entropy aligns with executive dysfunction literature.
- Minimal preprocessing: The approach requires less manual intervention compared to traditional linguistic preprocessing methods, making it more scalable and reproducible.

**Weaknesses:**

- Missing baselines and comparisons: The paper lacks comparison to previous computational methods for analyzing semantic fluency data. No baselines using simpler embeddings (e.g., Word2Vec, GloVe) or traditional NLP metrics are provided. Prior work like Linz et al. (2017) used word embeddings for similar tasks but isn't compared against.
- Limited theoretical grounding: While the authors claim semantic retrieval can be "understood as navigation through a multidimensional space" (Hills et al., 2015), this theoretical framework needs stronger support. The connection between observed metrics and established cognitive theories (e.g., clustering-switching models by Troyer et al., 1997) is underdeveloped.
- Interpretation of results lacks depth: The clinical findings (e.g., "greater spread, higher variability, increased entropy" in patient groups) are presented without sufficient discussion of whether these align with expected patterns from cognitive neuroscience literature. Are these results validating known theories or revealing new phenomena?
- Euclidean assumption: The authors acknowledge but don't address their "assumption of Euclidean dynamics" which "overlooks the anisotropic nature of embedding spaces" (citing Nickel & Kiela, 2017; Ethayarajh, 2019). This could significantly impact the validity of velocity and acceleration metrics.

**Questions:**

- Centroid computation: The distance-to-centroid shows lowest inter-model correlation. Could you elaborate on why this metric is particularly sensitive to model-specific geometry? How does collapsing repeated properties affect this measure?
- Category effects: The category-specific patterns differ between Italian and German datasets. Do these differences reflect linguistic/cultural variations or task administration differences? This needs deeper analysis.

---

> ### Author Response · Authors · 2025-11-23
> **Response to Reviewer xS4F (1)**
>
> Dear reviewer,
>
> Thank you for taking the time to read our work carefully. We appreciate the comments, and we will use them to improve the relevance and clarity of our theoretical and methodological framework. Below, we address your specific suggestions.
>
> > W1: Missing baselines and comparisons: The paper lacks comparison to previous computational methods for analyzing semantic fluency data. No baselines using simpler embeddings (e.g., Word2Vec, GloVe) or traditional NLP metrics are provided. Prior work like Linz et al. (2017) used word embeddings for similar tasks but isn't compared against.
>
> A-W1: We thank the reviewer for highlighting the importance of benchmarking our framework against established methods. To address this, we introduced a non-cumulative fastText (Mikolov, et al., 2018) baseline condition version of all metrics to directly replicate the traditional pairwise approach used in prior work like Linz et al. (2017). This baseline functions as a representative of static embeddings comparable to Word2Vec or GloVe. Additionally, we extend our experiments and compare all models against their non-cumulative versions. We further detailed these analyses in Table 2 and Appendix A.3.
>
> Our analysis revealed a distinct dissociation based on trajectory length. While traditional non-cumulative metrics yielded stronger effect sizes for the short word lists in the Italian and German datasets, our proposed cumulative transformer approach significantly outperformed non-cumulative baselines (including fastText) in the Neurodegenerative dataset. This indicates that while traditional pairwise methods may be sensitive when context is scarce, modeling the cumulative search history is critical for capturing the broader semantic structure and disorganized patterns found in clinical groups producing longer trajectories.
>
> > W2: Limited theoretical grounding: While the authors claim semantic retrieval can be "understood as navigation through a multidimensional space" (Hills et al., 2015), this theoretical framework needs stronger support. The connection between observed metrics and established cognitive theories (e.g., clustering-switching models by Troyer et al., 1997) is underdeveloped.
>
> A-W2: Thank you very much for your suggestion. We agree that the link between our metrics and established cognitive theories needs strengthening. We have rewritten the Introduction to explicitly frame our approach as an extension of the clustering/switching paradigm (Troyer et al., 1997) and foraging models (Hills et al., 2012). Specifically, we address it as follows in the Introduction, paragraph 2. We now posit that our kinematic metrics (velocity, acceleration) capture the "step-by-step granularity" of the search process, offering a continuous view that complements the binary nature of clustering and switching.
>
> > W3: Interpretation of results lacks depth: The clinical findings (e.g., "greater spread, higher variability, increased entropy" in patient groups) are presented without sufficient discussion of whether these align with expected patterns from cognitive neuroscience literature. Are these results validating known theories or revealing new phenomena?
>
> A-W3: We agreed on the need to deep into our clinical findings in light of current literature, and we address this now more clearly in the discussion section. We now further the interpretation of our clinical findings, interpreting high velocity/acceleration and entropy results in patients as signatures of executive dysfunction and diminished semantic space due to the loss of sensorimotor semantic traces to navigate. We have incorporated key literature to ground these interpretations in cognitive neuroscience literature (Speed et al., 2017; Fernandino et al., 2013; Birba et al., 2017). Regarding this, we expanded on how our results not only validate previous work but also reveal new characteristics of semantic navigation that can be used to better understand the cognitive profile of patients. This is addressed in the Discussion section, paragraph 2.
>
> > W4: Euclidean assumption: The authors acknowledge but don't address their "assumption of Euclidean dynamics" which "overlooks the anisotropic nature of embedding spaces" (citing Nickel & Kiela, 2017; Ethayarajh, 2019). This could significantly impact the validity of velocity and acceleration metrics.
>
> A-W4: We agree with you and the reviewers who raised this problem. We decided to address the embedding anisotropy by applying a ZCA-whitening transformation to our embeddings and recalculated the metrics to assess the robustness of group discrimination under more approximated isotropic conditions (Bell, 1997; Su et al., 2021; Zhuo et al., 2023). These results showed only small changes in group differences. We added this in the methods and limitations sections; the results are briefly discussed in the Appendix section A.4.

---

> ### Author Response · Authors · 2025-11-23
> **Response to Reviewer xS4F (2)**
>
> > Q1: Centroid computation: The distance-to-centroid shows lowest inter-model correlation. Could you elaborate on why this metric is particularly sensitive to model-specific geometry? How does collapsing repeated properties affect this measure?
>
> A-Q1: We hypothesize that the distance-to-centroid metric exhibits the lowest inter-model correlation because centroids are global statistics that are sensitive to the specific geometry of an embedding space. As noted by Ethayarajh (2019), transformer embeddings frequently exhibit anisotropy, meaning word representations occupy a narrow cone in the vector space rather than being uniformly distributed. Furthermore, centroids are susceptible to 'rogue dimensions' (Timkey & van Schijndel, 2021), where a limited number of dimensions dominate the variance, yielding unique, model-specific distance profiles. Despite this sensitivity, correlations remain relatively strong (0.5 - 0.8). The most significant divergence was observed in the Qwen3 model (0.6B); this outlier status likely stems from its causal training pipeline (Lee et al., 2025) and parameter scale, which contrast with bidirectional architectures (Zhang et al., 2024) or proprietary formulations. We added this to the result section 3.5.
>
> > Q2: Category effects: The category-specific patterns differ between Italian and German datasets. Do these differences reflect linguistic/cultural variations or task administration differences? This needs deeper analysis.
>
> A-Q2: Thank you for bringing this up; it allowed us to clarify dismissed elements in the discussion. First, we clarify that data acquisition protocols were identical for both languages, as described by Kremer & Baroni, 2011, ruling out administration artifacts. We expanded the discussion to attribute these differences to the interplay between linguistic structure and semantic representation, citing evidence on how psycholinguistic variables are encoded differently across languages. We now address this more explicitly in the Discussion section, paragraph 4.

---

> > ### Author Response · Authors · 2025-11-23
> > **References**
> >
> > Bell, A. J., & Sejnowski, T. J. (1997). The “independent components” of natural scenes are edge filters. Vision Research, 37(23), 3327–3338. https://doi.org/10.1016/S0042-6989(97)00121-1
> >
> > Birba, A., García-Cordero, I., Kozono, G., Legaz, A., Ibáñez, A., Sedeño, L., & García, A. M. (2017). Losing ground: Frontostriatal atrophy disrupts language embodiment in Parkinson's and Huntington's disease. Neuroscience & Biobehavioral Reviews, 80, 673–687. https://doi.org/10.1016/j.neubiorev.2017.07.011
> >
> > Ethayarajh, K. (2019). How contextual are contextualized word representations? Comparing the geometry of BERT, ELMo, and GPT-2 embeddings. In K. Inui, J. Jiang, V. Ng, & X. Wan (Eds.), Proceedings of the 2019 Conference on Empirical Methods in Natural Language Processing and the 9th International Joint Conference on Natural Language Processing (EMNLP-IJCNLP) (pp. 55–65). Association for Computational Linguistics. https://doi.org/10.18653/v1/D19-1006
> >
> > Fernandino, L., Conant, L. L., Binder, J. R., Blindauer, K., Hiner, B., Spangler, K., & Desai, R. H. (2013). Parkinson’s disease disrupts both automatic and controlled processing of action verbs. Brain and Language, 127(1), 65–74.
> >
> > Hills, T. T., Jones, M. N., & Todd, P. M. (2012). Optimal foraging in semantic memory. Psychological Review, 119(2), 431–440. https://doi.org/10.1037/a0027373
> >
> > Kremer, G., & Baroni, M. (2011). A set of semantic norms for German and Italian. Behavior Research Methods, 43(1), 97-109.
> >
> > Lee, J., Dai, Z., Ren, X., Chen, B., Cer, D., Cole, J. R., Hui, K., Boratko, M., Kapadia, R., Ding, W., Luan, Y., Duddu, S. M. K., Abrego, G. H., Shi, W., Gupta, N., Kusupati, A., Jain, P., Jonnalagadda, S. R., Chang, M.-W., & Naim, I. (2024). Gecko: Versatile text embeddings distilled from large language models. arXiv. https://doi.org/10.48550/arXiv.2403.20327
> >
> > Linz, N., Tröger, J., Alexandersson, J., & König, A. (2017). Using Neural Word Embeddings in the Analysis of the Clinical Semantic Verbal Fluency Task. In Proceedings of the 12th International Conference on Computational Semantics (IWCS) — Short Papers (pp. 1–7). The Association for Computer Linguistics. https://doi.org/10.18653/v1/W17-6926
> >
> > Mikolov, T., Grave, E., Bojanowski, P., Puhrsch, C., & Joulin, A. (2018). Advances in Pre-Training Distributed Word Representations. In Proceedings of the International Conference on Language Resources and Evaluation (LREC 2018).
> >
> > Speed, L. J., van Dam, W. O., Hirath, P., Vigliocco, G., & Desai, R. H. (2017). Impaired comprehension of speed verbs in Parkinson’s disease. Journal of the International Neuropsychological Society, 23(5), 412–420.
> >
> > Su, J., Cao, J., Liu, W., & Ou, Y. (2021). Whitening sentence representations for better semantics and faster retrieval. arXiv. https://doi.org/10.48550/arXiv.2103.15316
> >
> > Timkey, W., & van Schijndel, M. (2021). All bark and no bite: Rogue dimensions in transformer language models obscure representational quality. In M.-F. Moens, X. Huang, L. Specia, & S. W.-t. Yih (Eds.), Proceedings of the 2021 Conference on Empirical Methods in Natural Language Processing (pp. 4527–4546). Association for Computational Linguistics. https://doi.org/10.18653/v1/2021.emnlp-main.372
> >
> > Troyer, A. K., Moscovitch, M., & Winocur, G. (1997). Clustering and switching as two components of verbal fluency: Evidence from younger and older healthy adults. Neuropsychology, 11(1), 138–146. https://doi.org/10.1037/0894-4105.11.1.138
> >
> > Zhang, Y., Li, M., Long, D., Zhang, X., Lin, H., Yang, B., Xie, P., Yang, A., Liu, D., Lin, J., Huang, F., & Zhou, J. (2025). Qwen3 embedding: Advancing text embedding and reranking through foundation models (Version 3). arXiv. https://doi.org/10.48550/arXiv.2506.05176
> >
> > Zhuo, W., Sun, Y., Wang, X., Zhu, L., & Yang, Y. (2023). WhitenedCSE: Whitening-based contrastive learning of sentence embeddings. In A. Rogers, J. Boyd-Graber, & N. Okazaki (Eds.), Proceedings of the 61st Annual Meeting of the Association for Computational Linguistics (Volume 1: Long Papers) (pp. 12135–12148). Association for Computational Linguistics. https://doi.org/10.18653/v1/2023.acl-long.677

---

### Author Response · Authors · 2025-12-02
**Thank you for your reviews**

We thank the reviewers for their insightful comments, which help us substantially improve our work. In the manuscript, all modifications are highlighted in red for ease of review. We summarize the main changes below:

* Introduction: We clarified the theoretical foundations of our method, emphasizing the connection between cognitive and computational approaches to semantic search.
* Methods (theoretical link): We refined the psychological interpretation of our metrics and clarified how they relate to our methodological assumptions.
* Methods (data characteristics): We added Table 1 to clearly describe the datasets, aiding interpretation of the results.
* Appendix (benchmarking): We added comparisons against a classic fastText baseline and benchmarked all models using both non-cumulative and cumulative embeddings.
* Appendix (anisotropy): We applied ZCA-whitening to control for embedding anisotropy and re-computed all metrics to assess robustness.
* Discussion: We revised the discussion to better interpret our results in terms of semantic navigation rather than group classification. We also clarified why the Italian and German datasets, despite identical protocols, yield different patterns, underscoring cultural and linguistic influences on semantic memory.
* Limitations and generalizability: We clarified the generalizability of our approach and explicitly outlined its main limitations.

---

### Meta-Review · Area_Chair_nedp · 2026-01-05

**Summary:**

This paper proposes a framework to quantify human semantic navigation during concept production using trajectory-based metrics in transformer embedding spaces. It represents sequential concept generation (e.g., verbal fluency or property listing tasks) as a path through semantic space and computes geometric and dynamical metrics—distance to next, entropy, velocity, acceleration, and distance to centroid—to characterize this navigation. The method is applied to four datasets (neurodegenerative, swear-word fluency, and Italian/German property listing), showing that these metrics distinguish clinical groups, semantic categories, and languages. Results are robust across embedding models (OpenAI, Google, Qwen). The authors argue this approach bridges cognitive modeling and learned representations, offering potential applications in clinical and cross-linguistic research.

Reviewers agreed the approach is interesting,  however, they had issues with its mathematical and theoretical grounding, comparisons to prior methods, and how the work is being situated within the broader community.

The authors made substantial revisions to address the reviewers concerns including: adding baselines and benchmarking, addressing anisotropy with ZCA-whitening, rewriting Introduction and Discussion sections, adding clearer theoretical grounding, and improving figure readability.

**Reviewer Concerns:**

1.  No comparison to simpler embeddings (Word2Vec, GloVe, fastText) or traditional NLP metrics; prior work like Linz et al. (2017) not compared against.

    The authors dded a non-cumulative fastText baseline to replicate traditional pairwise approaches; they also extended experiments to compare all models against their non-cumulative versions, finding that cumulative transformer approach outperformed non-cumulative baselines for longer trajectories (clinical data), while traditional methods worked better for short word lists

2. Weak connection between metrics and established cognitive theories (e.g., clustering/switching models by Troyer et al., 1997); psychological interpretation of metrics underspecified.

    The authors rewrote Introduction to explicitly frame approach as extension of clustering/switching paradigm and foraging models, added clearer cognitive interpretations for each metric, positioned kinematic metrics as capturing "step-by-step granularity" that complements binary clustering/switching measures, and linked metrics to executive functions, working memory, and inhibition

3. The  proposed framework assumes Euclidean dynamics despite embeddings being anisotropic and non-Euclidea

The authors    applied ZCA-whitening transformation to embeddings to control for anisotropy and recalculated all metrics under more isotropic conditions; their results showed only small changes in group differences, demonstrating robustness

4. Clinical findings (greater spread, higher variability, increased entropy in patients) were presented without sufficient discussion of alignment with cognitive neuroscience literature.

    The authors expanded their Discussion section  to interpret findings as signatures of executive dysfunction, incorporated key literature on sensorimotor semantic traces and diminished semantic space in neurodegeneration, explained high velocity/acceleration and entropy as reflecting disorganized search patterns, and clarified whether results validate known theories or reveal new phenomena

5. Using non-causal encoder models means token B representation has access to token C in "A B C", which is cognitively implausible.

The authors clarified that three models with different architectures were used (including causal Qwen3); they howed trajectory-level metrics were consistent across models despite architectural differences, dded model architecture details to Methods section, and demonstrated robustness across causal vs. bidirectional attention mechanisms

There were some minor quibbles about the density of their figures which they addressed.

**Reviewer Scores:**

I think the authors did a great job in addressing the comments, my impression is that scores, in particular the 2 would have gone up.

---

### Decision · Program_Chairs · 2026-01-26

Accept (Poster)